# Mitigating Hallucination Through Theory-Consistent Symmetric Multimodal Preference Optimization

**Wenqi Liu**[1]    **Xuemeng Song**[2†]    **Jiaxi Li**[3]    **Yinwei Wei**[1]
**Na Zheng**[4]    **Jianhua Yin**[1†]    **Liqiang Nie**[5]

[1]Shandong University  [2]Southern University of Science and Technology  [3]University of Georgia
[4]National University of Singapore  [5]Harbin Institute of Technology (Shenzhen)
liuwq_bit@outlook.com, sxmustc@gmail.com, jhyin@sdu.edu.cn

## Abstract

Direct Preference Optimization (DPO) has emerged as an effective approach for mitigating hallucination in Multimodal Large Language Models (MLLMs). Although existing methods have achieved significant progress by utilizing vision-oriented contrastive objectives for enhancing MLLMs' attention to visual inputs and hence reducing hallucination, they suffer from non-rigorous optimization objective function and indirect preference supervision. To address these limitations, we propose a **Sym**metric **M**ultimodal **P**reference **O**ptimization (SymMPO), which conducts symmetric preference learning with direct preference supervision (i.e., response pairs) for visual understanding enhancement, while maintaining rigorous theoretical alignment with standard DPO. In addition to conventional ordinal preference learning, SymMPO introduces a preference margin consistency loss to quantitatively regulate the preference gap between symmetric preference pairs. Comprehensive evaluation across five benchmarks demonstrate SymMPO's superior performance, validating its effectiveness in hallucination mitigation of MLLMs. Our codes are available at https://github.com/Liuwq-bit/SymMPO.

## 1   Introduction

Recently, Large Language Models (LLMs) have demonstrated remarkable success in natural language understanding and generation tasks [1, 2]. By integrating visual encoders and cross-modal alignment modules to LLMs, Multimodal Large Language Models (MLLMs) [3, 4, 5] have extended capabilities of LLMs to the multimodal field, enabling applications such as visual question answering (VQA), image captioning, and multimodal dialogue systems. Despite their impressive performance, MLLMs often suffer from hallucination problems [6, 7], generating outputs inconsistent with the given image input or not relevant to the input textual prompt [8].

Due to its impressive performance in improving response quality by aligning LLMs with human preferences, i.e., guiding LLMs to generate outputs humans would judge as better (e.g., more safe or contextually appropriate), Direct Preference Optimization (DPO) [9] has been adapted to address hallucination issues in MLLMs [10, 11, 12, 13]. Existing methods [10, 14] first construct a preference pair consisting of a hallucination-free response $y_w$ and a hallucinated response $y_l$ for a given multimodal input, including an image $m$ and a textual prompt $x$ that specifies the generation task (e.g., VQA), and then perform DPO-based response-oriented preference learning (see Figure 1).To strengthen MLLMs' attention to visual inputs, recent studies further incorporate a DPO-based vision-oriented preference learning component [15, 16, 17, 18, 19], as shown in Figure 1. This component leverages contrastive triplet pairs $(m_w, x, y_w)$ and $(m_l, x, y_w)$ with only images varied, and aims to

---

[†]Corresponding authors.

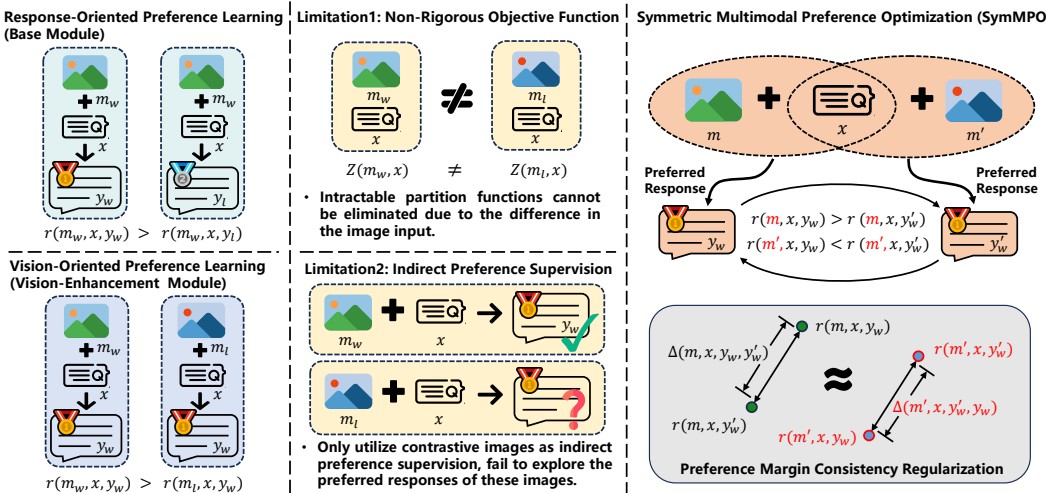

Figure 1: Existing methods focus on response- and vision-oriented preference learning (left), where the former serves as the base module, while the latter is optional, specifically for vision understanding enhancement. $r(m, x, y)$ represents the reward for generating response $y$ given the input $(m, x)$. However, current methods face two key limitations: non-rigorous objective function and indirect preference supervision (middle). To address these, we propose SymMPO, which leverages symmetric pairwise preference optimization using contrastive images and their preferred responses, alongside preference margin consistency regularization designed specifically for symmetric optimization (right).

make the likelihood of generating $y_w$ given $(m_w, x)$ higher than that given $(m_l, x)$. Here, $m_l$ is a modified version of $m_w$, obtained by either applying transformations to $m_w$ or adding noise to it.

While current methods have demonstrated promising results in MLLM hallucination mitigation by incorporating vision-oriented preference learning, they suffer from two key limitations.

**Limitation 1. Non-Rigorous Objective Function.** For deriving the objective function, existing works directly replace the shared image input $m$ in the pairwise likelihoods of the response-oriented preference learning objective function with its counterparts (i.e., $m_w$ and $m_l$) without rigorous mathematical justification. However, through a careful analysis of the standard multimodal DPO loss derivation, we find that the partition functions $Z(m_w, x)$ and $Z(m_l, x)$—which involve summing over all possible responses and are computationally expensive to estimate in the implicit reward formulation—cannot be directly canceled out in the loss function when the image inputs differ (i.e., $m_w \neq m_l$). In response-oriented preference learning, these partition functions naturally vanish due to their shared input $m$. However, this cancellation does not hold for vision-oriented preference learning, where the contrastive image inputs $m_w$ and $m_l$ are inherently distinct. Nevertheless, existing methods directly assume their eliminability, which results in a misaligned objective function that deviates from the theoretical formulation, ultimately limiting model performance [20]. A rigorous mathematical analysis regarding partition functions is provided in Appendix B.

**Limitation 2. Indirect Preference Supervision.** As shown in Figure 1, existing methods conduct DPO with two image-text-response triplets with fixed prompt and response but contrastive images for vision-oriented preference learning. This design fundamentally relies on contrastive images for preference alignment, which misaligns with DPO's core principle of reward formulation via paired responses (i.e., preferred vs. less preferred answers). This misalignment fails to explicitly model the direct preference relationship between accurate and hallucinated responses conditioned on the given image, leading to suboptimal visual understanding capabilities. In fact, we can additionally construct a preferred response for the contrastive image $m_l$ using the same prompt for $m_w$. Since $m_w$ and $m_l$ share high similarity, their corresponding preferred responses naturally exhibit strong semantic alignment with subtle variations—making them ideal hard negatives for each other. Such contrastive response pairs with minor differences compel the model to perform more thorough multimodal interpretation, thereby effectively reducing hallucinations.

To address these limitations, we propose **Sym**metric **M**ultimodal **P**reference **O**ptimization (SymMPO), to reduce hallucinations in MLLMs. Unlike existing methods that solely rely on contrastive images

to conduct DPO-based vision-oriented preference alignment, SymMPO additionally introduces the preferred response of the contrastive image given the same prompt of original image-prompt-response triplet, enabling vision-oriented preference learning with direct preference supervision (i.e., contrastive response pairs). Specifically, we design a symmetric preference optimization formulation, which simultaneously maximizes the likelihood of ranking $y_m$ over $y'_m$ given $(m, x)$ and that of ranking $y'_m$ over $y_m$ given $(m', x)$, where $m'$ is an image semantically similar to $m$ (identified via CLIP visual similarity), and $y'_m$ is the preferred response for $m'$ given $x$. The symmetric optimization compels the model to thoroughly differentiate between contrastive responses under varying visual contexts. By aligning with standard DPO in terms of using contrastive responses and fixed inputs $(m, x)$, our method naturally eliminates the two partition functions in the reward formulation, yielding a rigorous objective. Further, beyond conventional ordinal preference ranking optimization (i,e., $y_w \succ y'_w$), SymMPO introduces preference margin consistency regularization that enforces consistency in magnitude of preference gaps between contrastive responses. Additionally, SymMPO integrates the established response-oriented preference learning loss due to its remarkable performance in ensuring response quality, and introduces a cost-effective caption-anchored pipeline for constructing response preference pairs, facilitating efficient training and evaluation of our model.

Our main contributions can be summarized in three key aspects:

- We identify two key limitations in existing DPO-based multimodal preference optimization methods for MLLMs: non-rigorous objective function and indirect preference supervision for vision-oriented preference learning. For the non-rigorous objective function, we also provide detailed mathematical derivation in Appendix B.

- We propose SymMPO that effectively leverages preferred responses for contrastive images, supporting direct vision-oriented preference learning with a theory-consistent symmetric objective function. Additionally, we design a preference margin consistency regularization to quantitatively constrain the preference gap between symmetric response pairs, enabling more precise contrastive preference learning.

- We empirically validate the superiority of SymMPO over existing approaches through comprehensive experiments on four well-established MLLM hallucination benchmarks, as well as one benchmark designed to evaluate the ability of MLLMs to answer question based on images.

## 2 Preliminary

To facilitate our method presentation, we first review Proximal Policy Optimization (DPO's foundation), standard DPO, and existing multimodal DPO learning paradigm.

### 2.1 RLHF with Bradley-Terry Reward Model

Reinforcement Learning with Human Feedback (RLHF) [21, 22] has become a widely adopted approach for aligning LLMs with human preferences and expectations, enabling the generation of responses that better meet user needs. Among the various RLHF methods, Proximal Policy Optimization (PPO) [23] stands out as one of the most commonly used methods, primarily due to its stability and efficiency in policy optimization. PPO operates in two main stages: first, it trains a reward model that serves as a proxy for human feedback; second, it optimizes a policy model based on this reward model to guide the LLM toward generating responses that more effectively align with human expectations and preferences.

Specifically, to derive the reward model, PPO employs the widely adopted Bradley-Terry model [24], which maximizes the likelihood of the preferred (winning) response $y_w$ being ranked higher than the less preferred (losing) response $y_l$ for a given input $x$. This is formalized as follows:

$$P_{BT}(y_w \succ y_l|x) = \sigma(r_\phi(x, y_w) - r_\phi(x, y_l)) = \frac{\exp(r_\phi(x, y_w))}{\exp(r_\phi(x, y_w)) + \exp(r_\phi(x, y_l))}, \quad (1)$$

where $\phi$ represents the parameters of the reward model. $\sigma(\cdot)$ denotes the sigmoid function.

Based on the trained reward model $r_\phi$, PPO optimizes the policy model as follows:

$$\max_{\pi_\theta} \mathbb{E}_{x \sim \mathcal{D}, y \sim \pi_\theta(\cdot|x)} \Big[ r_\phi(x, y) - \beta D_{KL} \big( \pi_\theta(\cdot|x) \| \pi_{ref}(\cdot|x) \big) \Big], \quad (2)$$

where $x \in \mathcal{D}$ represents the input text sampled from dataset $\mathcal{D}$, and $y$ denotes the response generated by the current policy model $\pi_\theta$ parameterized with $\theta$. $\pi_{ref}$ is a fixed reference model used for constraining the policy model through Kullback-Leibler (KL) divergence $D_{KL}(\cdot\|\cdot)$. This objective function ensures the policy model $\pi_\theta$ progressively aligns with human preferences captured by the reward model $r_\phi$, while maintaining generation stability and preventing mode collapse. The hyper-parameter $\beta$ controls the strength of the KL penalty.

## 2.2 Standard Direct Preference Optimization

Although the PPO framework demonstrates effective performance in RLHF applications, it involves a separate additional reward model training process, limiting the training efficiency [25]. To address this, DPO [9] was introduced to eliminate the need for explicit reward modeling, while preserving remarkable effectiveness in aligning LLMs' output with human preferences. In particular, DPO introduces an implicit reward formulation that effectively integrates the probabilities yielded by the policy model and reference model, enabling the direct optimization of the policy model through establishing a direct mapping between policy parameters and human preferences via the Bradley-Terry model. In particular, the implicit reward formulation integrates the probabilities yielded by the policy model and reference model as follows:

$$r(x, y) = \beta \log \frac{\pi_\theta(y|x)}{\pi_{ref}(y|x)} + \beta \log Z(x), \qquad (3)$$

where $Z(x) = \sum_y \pi_{ref}(y|x) \exp\left(\frac{1}{\beta} r(x, y)\right)$ represents an intractable partition function, and $\beta$ is a hyper-parameter that controls the deviation from the reference model.

Although the implicit reward involves an intractable partition function, substituting it into the Bradley-Terry model (Equation 1) and applying the negative logarithmic transformation allows us to derive a DPO objective function free of any intractable terms, as follows:

$$\mathcal{L}_{DPO} = -\mathbb{E}_{(x.y_w,y_l)\sim\mathcal{D}}\left[ \log \sigma\left( \beta \log \frac{\pi_\theta(y_w|x)}{\pi_{ref}(y_w|x)} - \beta \log \frac{\pi_\theta(y_l|x)}{\pi_{ref}(y_l|x)} \right) \right], \qquad (4)$$

where, consistent with Equation 1, $y_w$ and $y_l$ denote the preferred and less preferred response, respectively, for given input text $x$.

## 2.3 Multimodal Direct Preference Optimization

Existing DPO-based multimodal preference optimization methods typically involve two key components: response-oriented preference learning [14, 11] and vision-oriented preference learning [15, 19]. While the former is universally adopted for response preference alignment, the latter is optional for enhancing visual interpretation.

**Response-oriented Preference Learning.** The standard DPO is designed for aligning LLMs to human preferences based on pairs of preferred and less preferred responses (i.e., $y_w$ and $y_l$). To align MLLMs with human preferences, existing methods typically directly extend the DPO objective by including an image condition $m$ as follows,

$$\mathcal{L}_{DPO_m} = -\mathbb{E}_{(x,m,y_w,y_l)\sim\mathcal{D}}\left[ \log \sigma\left( \beta \log \frac{\pi_\theta(y_w|m,x)}{\pi_{ref}(y_w|m,x)} - \beta \log \frac{\pi_\theta(y_l|m,x)}{\pi_{ref}(y_l|m,x)} \right) \right]. \qquad (5)$$

This objective maximizes the likelihood of the preferred (winning) response $y_w$ being ranked higher than the less preferred (losing) response $y_l$ for a given multimodal input $(m, x)$. Intuitively, it expects the model to capture the preference distinctions based on the overall multimodal input.

**Vision-oriented Preference Learning.** Despite its effectiveness, the overall response-oriented preference learning cannot guarantee MLLMs to accurately interpret the visual content. Therefore, recent studies incorporate a vision-oriented contrastive mechanism to enhance the MLLMs' visual understanding and hence generate more accurate responses with less hallucination. Specifically, in addition to $\mathcal{L}_{DPO_m}$, they introduce a vision-oriented contrastive objective $\mathcal{L}_{VCO}$ defined as:

$$\mathcal{L}_{VCO} = -\mathbb{E}_{(x,m_w,m_l,y_w)\sim\mathcal{D}}\left[ \log \sigma\left( \beta \log \frac{\pi_\theta(y_w|m_w,x)}{\pi_{ref}(y_w|m_w,x)} - \beta \log \frac{\pi_\theta(y_w|m_l,x)}{\pi_{ref}(y_w|m_l,x)} \right) \right], \qquad (6)$$

where $m_w$ represents the original image sampled from training dataset, and $m_l$ denotes a contrastive variant of $m_w$ generated through certain image manipulation operations (e.g., cropping and rotation). As can be seen, unlike $\mathcal{L}_{DPO_m}$ (see Equation 5), this objective takes the image-text pairs with fixed text (i.e., $x$) and varied images (i.e., $m_w$ and $m_l$) as the input condition, and adopts the same response (i.e., $y_w$) for both terms. By making the image condition the only variable, this objective encourages the model to learn preference distinctions based solely on visual information, thereby enhancing the MLLM's understanding of visual input.

# 3 Symmetric Multimodal Preference Optimization

To address the non-rigorous objective function and indirect preference supervision of existing vision-enhanced multimodal preference optimization methods, we propose SymMPO that conducts symmetric pairwise preference learning and preference margin consistency regularization, to enhance visual understanding and reduce MLLM hallucinations. Notably, SymMPO also incorporates the response-oriented preference learning, a well-established universal component for ensuring response quality. Here, we omit its details as they are covered in Section 2.3.

**Pair-wise Preference Learning.** Unlike existing vision-enhanced multimodal preference alignment methods that use indirect preference supervision (i.e., contrastive images), we seek direct preference supervision by further generating a preferred response $y'_w$ for the contrastive image $m'$ given the same prompt $x$. $m'$ is designed to differ only subtly from $m$, thereby leading to similar preferred responses (i.e., $y_w$ and $y'_w$) with minor variations. This intrinsic relationship naturally establishes $y_w$ and $y'_w$ as a contrastive pair, for enhancing the MLLMs' visual understanding capabilities. Towards comprehensive preference learning, SymMPO designs the following symmetric pairwise reward modeling function with the Bradley-Terry model:

$$
\begin{aligned}
&P_{BT}(y_w \succ y'_w | m, x) \wedge P_{BT}(y'_w \succ y_w | m', x) \\
&= \sigma\big(r(m, x, y_w) - r(m, x, y'_w)\big) \cdot \sigma\big(r(m', x, y'_w) - r(m', x, y_w)\big).
\end{aligned} \tag{7}
$$

This function essentially jointly models two pair-wise rewards, each aims to maximize the likelihood that the original preferred response ranks higher than the hard negative (contrastive) response for the given multimodal input. This encourages MLLMs to better interpret the given input image and reduce hallucinations. Then following DPO, we derive the optimization loss by applying the negative logarithmic transformation to the pairwise reward modeling function as follows,

$$
\begin{aligned}
\mathcal{L}_{Pair} =& - \mathbb{E}_{(x,m,m',y_w,y'_w) \sim \mathcal{D}} \Big[ \log \sigma\big(r(m, x, y_w) - r(m, x, y'_w)\big) + \log \sigma\big(r(m', x, y'_w) - r(m', x, y_w)\big) \Big] \\
=& - \mathbb{E}_{(x,m,m',y_w,y'_w) \sim \mathcal{D}} \Big[ \log \sigma\Big(\beta \log \frac{\pi_\theta(y_w|m,x)}{\pi_{ref}(y_w|m,x)} - \beta \log \frac{\pi_\theta(y'_w|m,x)}{\pi_{ref}(y'_w|m,x)}\Big) \\
& + \log \sigma\Big(\beta \log \frac{\pi_\theta(y'_w|m',x)}{\pi_{ref}(y'_w|m',x)} - \beta \log \frac{\pi_\theta(y_w|m',x)}{\pi_{ref}(y_w|m',x)}\Big) \Big].
\end{aligned} \tag{8}
$$

Since the above objective function aligns with standard DPO by using response pairs as direct preference supervision, SymMPO naturally cancels out the intractable partition functions that share the same multimodal inputs, leading to a rigorous objective function.

**Preference Margin Consistency Regularization.** As shown in Equation 7, the Bradley-Terry model only separately constrains $r(m, x, y_w) > r(m, x, y'_w)$ and $r(m', x, y'_w) > r(m', x, y_w)$. However, since $m'$ is a perturbed variant of $m$, the preference margin between $r(m, x, y_w)$ and $r(m, x, y'_w)$ should be similar to that between $r(m', x, y'_w)$ and $r(m', x, y_w)$ for near-identical inputs (i.e., $(m, x)$ and $(m', x)$). Thus, beyond the traditional ordinal preference learning, we incorporate a preference margin consistency regularization, which quantitatively regulate the response preference margin across contrastive images as follows:

$$
\begin{cases}
\mathcal{L}_{Margin} = \mathbb{E}_{(x,m,m',y_w,y'_w) \sim \mathcal{D}} \Big(\Delta(m, x, y_w, y'_w) - \Delta(m', x, y'_w, y_w)\Big)^2, \\
\Delta(m, x, y_w, y'_w) = r(m, x, y_w) - r(m, x, y'_w) = \log \frac{\pi_\theta(y_w|m,x)}{\pi_{ref}(y_w|m,x)} - \log \frac{\pi_\theta(y'_w|m,x)}{\pi_{ref}(y'_w|m,x)},
\end{cases} \tag{9}
$$

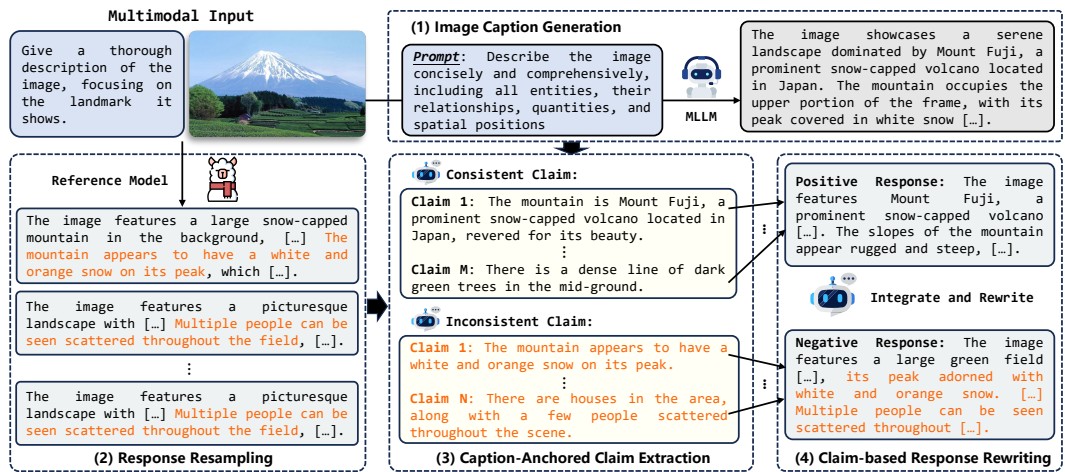

Figure 2: The caption-anchored response preference pair construction pipeline.

where $\Delta(m, x, y_w, y'_w)$ quantifies the preference margin between $y_w$ and $y'_w$ given $(m, x)$.

Moreover, to prevent the model from reducing the likelihood of the preferred response for optimizing the likelihood gap between preferred and less preferred responses, which can be harmful to preference alignment, we also adopt the anchored preference regularization [15, 19] as follows:

$$\mathcal{L}_{AncPO} = -\mathbb{E}_{(x,m,m',y_w,y'_w)\sim\mathcal{D}}\Big[\log\sigma\Big(\beta\log\frac{\pi_\theta(y_w|m,x)}{\pi_{ref}(y_w|m,x)} - \delta\Big) + \log\sigma\Big(\beta\log\frac{\pi_\theta(y'_w|m',x)}{\pi_{ref}(y'_w|m',x)} - \delta\Big)\Big],$$
(10)

where $\delta$ is the anchor that encourages the model to yield high likelihoods for preferred responses.

Ultimately, the complete optimization objective of SymMPO is formally defined as follows:

$$\mathcal{L}_{SymMPO} = \mathcal{L}_{DPO_m} + \lambda\mathcal{L}_{Pair} + \gamma\mathcal{L}_{Margin} + \eta\mathcal{L}_{AncPO},$$
(11)

where $\lambda$, $\gamma$, and $\eta$ are weighting hyper-parameters.

## 4 Experiment

In this section, we present our training data construction process, experimental setup and results.

### 4.1 Training Data Construction

In this work, we adopt the same set of 21.4k image-prompt pairs from TPO [26], which aggregates multiple public datasets, including VQA v2 [27], MSCOCO [28] and TextVQA [29]. To construct preference data for model optimization, we first generate contrastive responses $(y_w, y_l)$ for each image-prompt pair $(m, x)$ to compute $\mathcal{L}_{DPO_m}$. Subsequently, we create a contrastive image $m'$ with its corresponding positive response $y'_m$, forming contrastive triplets $(m, x, y_m)$ and $(m', x, y'_m)$ for calculating $\mathcal{L}_{Pair}$, $\mathcal{L}_{Margin}$, and $\mathcal{L}_{AncPO}$. This process essentially involves two key modules: preference response pair construction and contrastive image construction, while $y'_m$ for $m'$ can be easily obtained by the preference response pair construction module.

**Preference Response Pair Construction**. While previous works have proposed various methods for preference pair construction [13, 11], these approaches typically suffer from either costly API calls to commercial MLLMs (e.g., GPT-4V [4]) or substantial computational overhead with complex multi-stage processing. Therefore, we design a cost-effective Caption-Anchored Claim Extraction-and-Rewriting pipeline, as illustrated in Figure 2, comprising four efficient stages: 1) Image Caption Generation, 2) Response Resampling, 3) Caption-Anchored Claim Extraction, and 4) Claim-Based Response Rewriting. First, we use an open-source MLLM (e.g., Qwen2.5-VL-32B [30, 31]) to generate a detailed caption for the image. Next, multiple responses are generated from the reference model. Unlike prior approaches [11, 26], the third stage of our pipeline directly extracts atomic claims by performing fine-grained comparison between each response and the detailed image caption,

leveraging a high-performance LLM (e.g., DeepSeek-V3 [32]). This simplifies the process by removing the need for explicit claim extraction, claim-to-question conversion, and per-claim verification as required in previous methods [11, 26]. Instead, our approach only requires generating the image caption and designing a prompt that instructs the LLM to extract positive claims (consistent with the caption) and negative claims (inconsistent with it). Finally, an LLM (e.g., DeepSeek-V3) is further used as a rewrite model to generate positive and negative responses based on the extracted consistent and inconsistent claims, respectively.

**Contrastive Image Construction**. Unlike existing methods that generate the contrastive image $m'$ by applying transformations or noise to the original image $m$, we adopt an alternative construction approach. Specifically, we compute CLIP [33] embeddings for all images in the dataset and pair each image with its nearest neighbor based on cosine similarity. This ensures $m'$ and $m$ share high visual similarity while exhibiting subtle differences, making their positive responses $(y_m, y'_m)$ more challenging to distinguish, thereby enhancing symmetric pair-wise preference learning.

## 4.2 Experimental Setup.

**Benchmarks.** For evaluation, we adopt five established benchmarks: 1) **HallusionBench** [7] evaluates both language hallucination and visual illusion, featuring 346 images and 1,129 prompts. It employs GPT-4 [1] to compare MLLM outputs against ground truth, using three evaluation metrics: question-level accuracy (qAcc), figure-level accuracy (fAcc), and overall accuracy across all prompts (aAcc). For cost efficiency, we substitute the original GPT-4 [1] evaluator with DeepSeek-V3 [32]. 2) **Object-HalBench** [34] is a benchmark for evaluating common object hallucination in detailed image descriptions. Following [10] and [11], we use eight diverse prompts across 300 instances to ensure robust evaluation, reporting both response-level and mention-level hallucination rates. 3) **MMHal-Bench** [14] evaluates response informativeness and hallucination rates using GPT-4 to compare model outputs against human annotations for 96 images. The information score and hallucination rate serve as the evaluation metrics. 4) **AMBER** [35] provides 15k fine-grained annotations with well-designed prompts, enabling comprehensive evaluation across three key aspects: object existence, attribute accuracy, and relational correctness. We report the accuracy and F1 score for this discriminative evaluation. 5) **MMStar** [36] provides 1.5k image-prompt pairs for evaluating six core capabilities and 18 specific aspects of MLLMs, with results summarized in an overall performance score.

**Baselines.** For fair evaluation, we first adopt the following PPO/DPO-based baselines with publicly available pretrained weights: PPO-based method (LLaVA-RLHF [14]) and several DPO-based methods, including RLHF-V [10] (trained on Muffin-13B [37]), as well as POVID [38], HALVA [39], HA-DPO [13], RLAIF-V [11], TPO [26], OPA-DPO [19], and HSA-DPO [40] (all trained on LLaVA-1.5 [3]). To eliminate confounding factors from differing experimental settings and ensure a rigorous comparison, we also incorporate standard DPO [9] and mDPO [15] in our evaluation. Both methods are trained under the same experimental conditions as SymMPO, including identical training data, parameter configurations, and environment. All models (DPO, mDPO, and SymMPO) for rigorous comparison are trained for 2 epochs with a learning rate of 5e-6 and batch size of 64, using the following hyper-parameters: $\beta = 0.1$, $\delta = 0$, $\lambda = 0.5$, $\gamma = 1e-4$, and $\eta = 1.0$. The training is performed on 4 NVIDIA A100-40GB GPUs.

## 4.3 Main Results

Table 1 presents the performance comparison across five benchmarks with two model variants of LLaVA-1.5 [3]: LLaVA-1.5-7B and LLaVA-1.5-13B . From this table, we have following observations: (1) SymMPO outperforms existing methods across most evaluation metrics for both LLaVA-1.5-7B and LLaVA-1.5-13B, demonstrating its effectiveness in reducing hallucination in MLLMs. (2) The blue-highlighted rigorous comparisons show SymMPO's consistent superiority over DPO and mDPO across four benchmarks: HallusionBench, MMHal-Bench, AMBER, and MMStar. This advantage stems from our novel vision-oriented contrastive learning framework, which effectively leverages symmetric comparisons through a rigorous objective function derived from standard DPO, thereby enhancing the model's visual understanding. (3) Surprisingly, both mDPO and our SymMPO underperform DPO on Object-HalBench. This may stem from a misalignment between our preference data construction method and Object-HalBench's evaluation task. In our data construction pipeline, we use an open-source MLLM to generate image captions, which serve as references for extracting consistent/inconsistent claims and generating positive/negative responses. However,

Table 1: Main experimental results. The best and second-best results under the same experiment setting are highlighted in boldface and underlined, respectively.

| Model | Data Size | Feedback | HallusionBench | | | Object-HalBench | | MMHal-Bench | | AMBER | | MMStar |
|---|---|---|---|---|---|---|---|---|---|---|---|---|
| | | | qAcc↑ | fAcc↑ | aAcc↑ | Resp.↓ | Ment.↓ | Score↑ | Hall↓ | Acc↑ | F1↑ | Overall↑ |
| **Muffin-13B** [37] | ✗ | ✗ | 6.15 | 12.71 | 41.89 | 53.0 | 24.3 | 2.06 | 66.7 | 74.2 | 80.0 | 25.4 |
| +RLHF-V [10] | 1.4k | Human | 9.67 | 13.87 | 45.79 | 8.5 | 4.9 | 2.60 | 56.2 | 82.0 | 86.7 | 31.0 |
| **LLaVA-1.5-7B** [3] | ✗ | ✗ | 3.95 | 11.56 | 41.71 | 56.5 | 27.9 | 2.26 | 56.2 | 71.8 | 74.5 | 33.3 |
| +LLaVA-RLHF [14] | 122k | Self-Reward | 5.49 | 12.13 | 38.26 | 55.4 | 27.3 | 2.00 | 66.7 | 68.7 | 74.7 | 31.4 |
| +POVID [38] | 17k | GPT-4V | 7.03 | 9.53 | 43.31 | 35.9 | 17.3 | 2.28 | 56.2 | 78.6 | 81.9 | 34.4 |
| +HALVA [39] | 21.5k | GPT-4V | 5.49 | 11.27 | 42.42 | 49.1 | 24.6 | 2.14 | 60.4 | 78.0 | 83.5 | 32.3 |
| +HA-DPO [13] | 6k | GPT-4 | 5.49 | 11.56 | 42.16 | 44.9 | 21.8 | 1.97 | 61.5 | 74.2 | 78.0 | 32.6 |
| +RLAIF-V [11] | 74.8k | LLaVA-Next | 5.93 | 5.49 | 36.75 | 9.9 | 4.9 | 3.04 | 39.6 | 72.7 | 84.4 | 34.6 |
| +TPO [26] | 21.4k | LLaVA-Next | 7.03 | 11.27 | 41.62 | 5.0 | 4.7 | 2.76 | 42.7 | 82.2 | 87.2 | 34.2 |
| +OPA-DPO [19] | 4.8k | LLaVA-Next | 6.37 | 11.84 | 42.69 | 6.1 | 3.7 | 2.83 | 46.9 | 81.3 | 85.6 | 33.1 |
| **+DPO** [9] | 21.4k | DeepSeek-V3 | **7.25** | 7.80 | 40.21 | **12.9** | 8.8 | 2.44 | 49.0 | 71.3 | 82.6 | 33.4 |
| **+mDPO** [15] | 21.4k | DeepSeek-V3 | 6.81 | 9.53 | 42.78 | 19.9 | 10.1 | 2.71 | 50.0 | 80.6 | 86.3 | 34.2 |
| **+SymMPO (Ours)** | 21.4k | DeepSeek-V3 | **7.25** | **13.58** | **44.28** | 19.5 | 9.7 | **2.89** | 42.7 | 82.6 | 87.7 | **34.8** |
| **LLaVA-1.5-13B** [3] | ✗ | ✗ | 6.59 | 9.53 | 43.48 | 51.2 | 25.1 | 2.16 | 59.4 | 71.3 | 73.2 | 33.1 |
| +LLaVA-RLHF [14] | 122k | Self-Reward | 8.57 | 10.11 | 43.48 | 45.3 | 21.5 | 2.15 | 66.7 | 79.7 | 83.9 | 33.5 |
| +HALVA [39] | 21.5k | GPT-4V | 8.79 | 10.11 | 42.24 | 47.0 | 22.9 | 2.30 | 57.3 | 82.9 | 86.5 | 33.1 |
| +HSA-DPO [40] | 8k | GPT-4/4V | 6.15 | 8.95 | 41.62 | 5.4 | 2.9 | 2.55 | 50.0 | 79.8 | 82.8 | 33.7 |
| +OPA-DPO [19] | 4.8k | LLaVA-Next | 6.81 | 12.13 | 42.60 | 7.7 | 4.4 | 3.05 | 38.5 | 84.1 | 87.5 | 32.3 |
| **+DPO** [9] | 21.4k | DeepSeek-V3 | 10.32 | 10.69 | 39.50 | **15.4** | 8.5 | 2.65 | 45.8 | 69.2 | 84.6 | 33.0 |
| **+mDPO** [15] | 21.4k | DeepSeek-V3 | 9.23 | 10.69 | 39.85 | 20.9 | 10.8 | 2.93 | 43.8 | 83.8 | 88.8 | 35.0 |
| **+SymMPO (Ours)** | 21.4k | DeepSeek-V3 | **10.54** | **10.98** | **44.55** | 20.4 | 10.0 | **3.01** | 39.6 | 84.9 | 89.1 | **35.2** |

Table 2: Ablation studies with LLaVA-1.5-7B.

| Model | HallusionBench | | | Object-HalBench | | MMHal-Bench | | AMBER | | MMStar |
|---|---|---|---|---|---|---|---|---|---|---|
| | qAcc↑ | fAcc↑ | aAcc↑ | Resp.↓ | Ment.↓ | Score↑ | Hall↓ | Acc↑ | F1↑ | Overall↑ |
| **SymMPO** | **7.25** | **13.58** | 44.28 | 19.5 | **9.7** | **2.89** | **42.7** | **82.6** | **87.7** | 34.8 |
| w/o-$\mathcal{L}_{Pair}$ | 6.59 | 11.84 | 43.22 | **18.1** | 10.6 | 2.53 | 50.0 | 81.7 | 87.1 | 33.8 |
| w/o-$\mathcal{L}_{Margin}$ | 7.03 | 10.98 | **44.46** | 21.1 | 11.0 | 2.40 | 54.2 | 82.0 | 87.3 | 34.5 |
| w/o-$\mathcal{L}_{AncPO}$ | 6.81 | 11.84 | 40.83 | 21.6 | 11.6 | 2.39 | 59.4 | 79.5 | 87.4 | **36.2** |

these captions tend to focus more on object detection and scene overview rather than fine-grained visual details, thereby introducing noise about subtle visual properties in preference response pairs. While this noise has a limited impact on benchmarks that assess hallucination using straightforward discriminative questions (e.g., multiple-choice or yes/no questions), it poses a greater challenge for Object-HalBench, which directly evaluates response- and mention-level hallucination in detailed MLLM-generated descriptions. Despite this, SymMPO still outperforms mDPO, demonstrating stronger performance in mitigating hallucination in fine-grained descriptions.

## 4.4 Ablation Study

To evaluate the contribution of each key component in SymMPO, we introduce three variants: w/o-$\mathcal{L}_{Pair}$, w/o-$\mathcal{L}_{Margin}$, and w/o-$\mathcal{L}_{AncPO}$, where the pair-wise preference loss, the margin consistency regularization, and the anchored preference regularization are removed, respectively. Table 2 presents the ablation results across five benchmarks using LLaVA-1.5-7B as the backbone. As we can see, the complete SymMPO model outperforms all three variants on most metrics, demonstrating each component's essential contribution to SymMPO's effectiveness. Specifically, the results confirm: (1) the critical importance of pair-wise preference learning for eliminating MLLM hallucination; (2) the necessity of regulating quantitative preference margins between paired samples beyond qualitative preference alignment; and (3) the positive effect of anchored preference regularization in improving training stability by preventing log probability decline of positive responses during optimization.

## 4.5 Impact of Contrastive Image

To investigate the impact of contrastive images on our SymMPO, we also perform experiments on four alternative types of contrastive images using LLaVA-1.5-7B. (1) **Black**: Completely black images containing no visual information; (2) **Cropped**: Sub-images randomly cropped from original images, with half the original width and height; (3) **Noisy**: Original images corrupted with Gaussian noise ($\sigma = 0.8$); and (4) **Synthetic**: Inspired by the great success of image captioning and generation

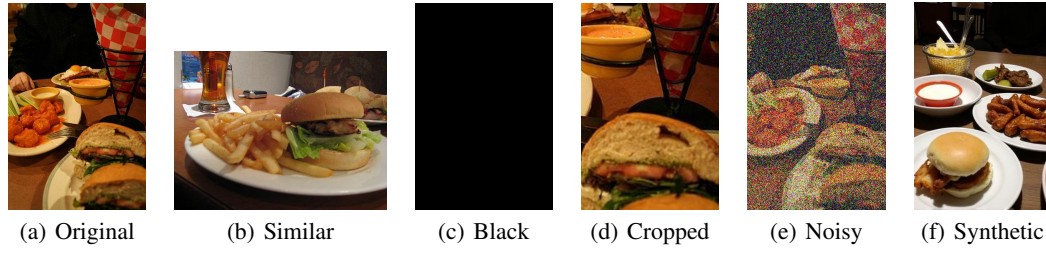

| (a) Original | (b) Similar | (c) Black | (d) Cropped | (e) Noisy | (f) Synthetic |

Figure 3: Samples of the original image and its related contrastive images.

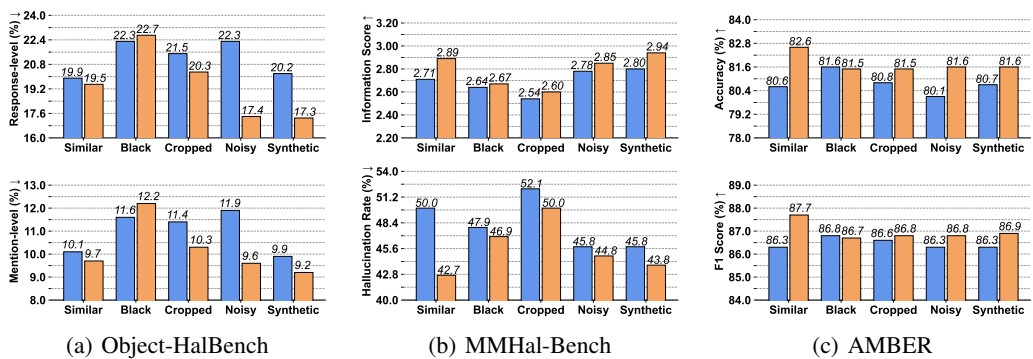

| (a) Object-HalBench | (b) MMHal-Bench | (c) AMBER |

Figure 4: Results of SymMPO and mDPO using different types of contrastive images ($\uparrow$/$\downarrow$: higher/lower is better). **Orange** represents SymMPO, and **blue** represents mDPO.

models, we synthesize the contrastive images by FLUX.1-dev [41] based on the LLaVA-1.5-7B-generated captions of original images. Figure 3 shows contrastive image examples, where "Similar" refers to our CLIP-based contrastive images used in main experiments.

Figure 4 shows performance comparison between our SymMPO and mDPO with different types of contrastive images on three mainstream benchmarks, including Object-HalBench, MMHal-Bench, and AMBER. From this figure, we make the following key observation: (1) SymMPO consistently outperforms mDPO across all types of contrastive images, except for black images. This demonstrates the robustness of our model in handling diverse contrastive inputs. The performance degradation of our SymMPO with black images may stem from their lack of visual information, which results in non-meaningful target responses. With insufficient meaningful signals, SymMPO struggles to perform effective preference optimization. In contrast, mDPO only relies on the varying visual inputs with the same response of the original image for visual understanding enhancement, making it less susceptible to this limitation. (2) Using Similar, Noisy or Synthetic images, demonstrates superior performance over using Black and Cropped images. This possible explanation is that the former three types of images preserve semantic similarities with the original images better, enabling more effective visual contrast through symmetric comparison. In contrast, black images suffer from severe information deficiency, while cropped images risk eliminating key visual elements relevant to the prompt, both factors weakening their effects in visual understanding enhancement.

## 5 Limitation

Despite its promising performance in mitigating hallucination of MLLMs, SymMPO has two limitations. (1) Due to constraints in our cost-effective and efficient preference data construction pipeline, SymMPO's performance on tasks that involve fine-grained visual understanding (e.g., detailed image description generation) remains limited (discussed in Section 4.3). (2) SymMPO introduces additional computational overhead as it requires the construction of preferred response for each contrastive image. Although we have designed a cost-effective caption-anchored response preference pair construction pipeline for reducing the computational cost, we acknowledge it as a limitation.

# 6    Conclusion

In this paper, we propose Symmetric Multimodal Preference Optimization (SymMPO), which effectively addresses the limitations of non-rigorous objective function and indirect preference supervision in existing vision-enhanced multimodal preference optimization methods. The key novelty lies in the symmetric pair-wise preference optimization formulation and preference margin consistency regularization. Extensive experiments conducted across five multimodal benchmarks validate the superiority of our SymMPO framework, while ablation studies confirm the necessity of each key component within the framework. Additional comparisons on contrastive image types highlight the robustness of our model with diverse types.

## Acknowledgments and Disclosure of Funding

This research is supported by the National Natural Science Foundation of China (Grants No. 62376137 and 62172261) and the Shandong Provincial Natural Science Foundation (Grant No. ZR2022YQ59), for which we express our heartfelt gratitude. We also deeply appreciate the support and assistance provided by the iLearn Laboratory at Shandong University during the course of this research.

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

# A Related Work

Existing approaches that apply DPO to align MLLMs fundamentally involves two components: (1) preference data construction and (2) optimization strategy. We accordingly present current multimodal DPO research from these two perspectives.

## A.1 Preference Data Construction

Exiting preference data construction approaches primarily fall into two categories: (1) comparative ranking-based methods and (2) hallucination correction-based methods.

The first category constructs preference data by directly evaluating the rankings of candidate responses. For example, CLIP-DPO [42] uses CLIP's image-text similarity metrics to rank candidate responses and selects pairs with large ranking gaps and similar lengths as preference data. Beyond this, AMP [12] leverages MLLMs of varying scales to generate multi-level preference data, assuming that responses from larger MLLMs should rank higher. Furthermore, BPO [43] distorts the input image and pairs the response generated from the distorted image (ranked lower) with the response from the original image (ranked higher) to create preference pairs. To eliminate confounding factors such as text style that hinder the model's ability to discern genuine trustworthiness differences within response pairs, RLAIF-V [11] introduces a deconfounded candidate response generation strategy. This strategy generates candidate responses through multiple sampling decoding trials with different random seeds while keeping the input prompt and decoding parameters constant.

The second category constructs preference data by detecting and correcting hallucinated content in MLLM responses. For example, RLHF-V [10] relies on human annotators to identify and rectify hallucinatory content, ensuring the creation of higher-quality preference datasets. In contrast, HA-DPO [13] bypasses human annotators by leveraging GPT-4 [1] to detect and correct hallucination using fine-grained visual annotations from the Visual Genome dataset [44]. Similarly, OPA-DPO [19] employs GPT-4V [4] to rectify hallucination but identifies them through direct image pair comparisons, eliminating the need for fine-grained annotations and offering greater flexibility. To reduce the high API costs associated with commercial LLMs, HSA-DPO [40] trains a specialized hallucination detection model using a sentence-level annotation dataset generated by GPT. Moreover, building on RLAIF-V [11], TPO [26] introduces a topic-level self-correction paradigm. This method works on identifying hallucination at the topic level through sub-sentence clustering, constructing topic-level preference pairs, and generating response preference pairs using a deconfounded topic-overwriting strategy—ensuring linguistic style consistency.

## A.2 Optimization Strategies

Apart from enhancing preference data construction, researchers have also explored various optimization strategies to improve MLLM preference alignment. For example, MPO [45] introduces a mixed preference optimization model, which effectively combines preference optimization techniques and conventional supervised fine-tuning. To mitigate the hallucination of MLLMs, AMP [12] introduces a multi-level direct preference optimization algorithm, enabling robust multi-level preference learning, while CHiP [18] introduces a cross-modal hierarchical DPO model involving two key optimizations: hierarchical textual preference optimization for capturing fine-grained textual preferences and visual preference optimization for cross-modal preference alignment. To address the gradient vanishing problem induced by off-policy data, OPA-DPO [19] proposes an adaptive mechanism that dynamically balances exploration and exploitation during learning. Furthermore, MIA-DPO [46] specifically targets hallucination reduction in multi-image scenarios, where MLLMs process multiple input images simultaneously through optimized cross-image attention mechanisms.

One key issue in improving MLLMs' reasoning ability is mitigating their over-reliance on textual prompts while enhancing visual content utilization. To address this issue, SymDPO [47] introduces a symbol demonstration direct preference optimization model for in-context learning, which strengthens MLLMs' visual understanding by replacing textual answers with random symbols, thereby forcing MLLMs to establish mappings between visual information and symbolic responses. For more general multimodal understanding contexts, several studies [15, 16, 17, 18, 19] focused on extending DPO with vision-oriented preference learning to improve MLLMs' visual signal interpretation. These approaches preserve DPO's structural framework while only introducing visual variation in the

contrastive image-prompt-response triplet pairs. For contrastive image generation, existing methods employ diverse strategies: mDPO [15] applies geometric transformations to original images, V-DPO [16] employs a generative model to replace key visual elements through image inpainting [48], MFPO [17], OPA-DPO [19] and CHiP [18] perform noise injection to original images, while CHiP [18] utilizes a forward diffusion process [49] for generating contrastive images.

Although current vision-enhanced multimodal preference alignment methods have demonstrated great progress in reducing MLLM hallucination, they exhibit two critical limitations: (1) their DPO-based objective function derivations lack mathematical rigor, as they overlook the fact that the intractable partition functions in multimodal scenarios with different vision inputs cannot be directly canceled; and (2) they fail to provide direct preference supervision for DPO-based visual understanding enhancement. To address these limitations, we propose Symmetric Multimodal Preference Optimization, which effectively utilize the corresponding preferred responses of contrastive images for optimizing the visual understanding capabilities of MLLMs.

## B  Impact of Partition Function in Multimodal Preference Optimization

Due to space limitations, Section 1 briefly argues that existing multimodal DPO methods [15, 16, 17, 18, 19] non-rigorously ignore two partition functions in their vision-oriented contrastive learning mechanisms. Here, we present a detailed analysis of these functions' roles in model optimization.

Specifically, according to the standard DPO formulation (Equation 3), the implicit reward of MLLMs can be expressed as follows:

$$
\begin{cases}
r(m, x, y) = \beta \log \dfrac{\pi_\theta(y|m, x)}{\pi_{ref}(y|m, x)} + \beta \log Z(m, x), \\
Z(m, x) = \sum_y \pi_{ref}(y|m, x) \exp\left(\dfrac{1}{\beta} r(m, x, y)\right),
\end{cases}
\tag{12}
$$

where $m$, $x$, and $y$ denote the input image, textual prompt, and corresponding response, respectively. $\pi_\theta$, $\pi_{ref}$, and $r$ are the policy model, reference model, and implicit reward function, respectively. $Z(m, x)$ is the partition function for the multimodal scenario, derived by incorporating an image variable into the single-modal partition function defined by standard DPO [9].

Adopting the mainstream vision-oriented preference learning paradigm that introduces additional contrastive images for preference alignment, we derive the following loss function by substituting the above implicit reward formulation into the Bradley-Terry model (Equation 1),

$$
\begin{aligned}
\mathcal{L}^*_{VCO} &= -\mathbb{E}_{(x, m_w, m_l, y_w) \sim \mathcal{D}} \Big[ \log \sigma\big(r(m_w, x, y_w) - r(m_l, x, y_w)\big) \Big] \\
&= -\mathbb{E}_{(x, m_w, m_l, y_w) \sim \mathcal{D}} \Big[ \log \sigma\Big(\beta \log \frac{\pi_\theta(y_w|m_w, x)}{\pi_{ref}(y_w|m_w, x)} - \beta \log \frac{\pi_\theta(y_w|m_l, x)}{\pi_{ref}(y_w|m_l, x)} \\
&\qquad\qquad + \beta \log Z(m_w, x) - \beta \log Z(m_l, x)\Big)\Big].
\end{aligned}
\tag{13}
$$

Existing methods directly cancel out $Z(m_w, x)$ and $Z(m_l, x)$ in Equation 6, which is apparently inappropriate based on the above rigorous derivation.

To better understand the role of partition functions in the preference learning process, we calculate the gradient of $\mathcal{L}^*_{VCO}$ with respect to $\theta$. To facilitate the gradient calculation, for each data instance sampled during training, $(x, m_w, m_l, y_w) \sim \mathcal{D}$, we first define:

$$
\begin{cases}
u = \beta \log \dfrac{\pi_\theta(y_w|m_w, x)}{\pi_{ref}(y_w|m_w, x)} - \beta \log \dfrac{\pi_\theta(y_w|m_l, x)}{\pi_{ref}(y_w|m_l, x)}, \\
c = \beta \log Z(m_w, x) - \beta \log Z(m_l, x).
\end{cases}
$$

where $u$ involves the policy model optimization, while $c$ does not. In fact, $c$ remains constant across different policy model parameters $\theta$ because $Z(m, x)$ is independent of $\theta$ in its calculation.

Table 3: Comparison of different learning rates on the performance of SymMPO

| SymMPO | HallusionBench | | | Object-HalBench | | MMHal-Bench | | AMBER | | MMStar |
|---|---|---|---|---|---|---|---|---|---|---|
| | qAcc↑ | fAcc↑ | aAcc↑ | Resp.↓ | Ment.↓ | Score↑ | Hall↓ | Acc↑ | F1↑ | Overall↑ |
| lr=5e-5 | **8.13** | 10.40 | 41.09 | 22.6 | 13.9 | 2.29 | 54.2 | 80.8 | 86.1 | 28.2 |
| lr=5e-6 | 7.25 | **13.58** | **44.28** | **19.5** | **9.7** | **2.89** | **42.7** | **82.6** | **87.7** | **34.8** |
| lr=5e-7 | **8.13** | 12.42 | 43.75 | 20.1 | 9.8 | 2.80 | 49.0 | 80.8 | 86.8 | 33.8 |

Table 4: Comparison of different $\lambda$ on the performance of SymMPO

| SymMPO | HallusionBench | | | Object-HalBench | | MMHal-Bench | | AMBER | | MMStar |
|---|---|---|---|---|---|---|---|---|---|---|
| | qAcc↑ | fAcc↑ | aAcc↑ | Resp.↓ | Ment.↓ | Score↑ | Hall↓ | Acc↑ | F1↑ | Overall↑ |
| $\lambda$=0.1 | 5.27 | 10.98 | 41.27 | 21.2 | 12.0 | 2.61 | 50.0 | 80.6 | 86.4 | 34.2 |
| $\lambda$=0.3 | **7.47** | 10.98 | 42.60 | 19.7 | 10.5 | **3.07** | **37.5** | 81.5 | 86.7 | 34.6 |
| $\lambda$=0.5 | 7.25 | **13.58** | **44.28** | 19.5 | 9.7 | 2.89 | 42.7 | **82.6** | **87.7** | **34.8** |
| $\lambda$=0.7 | 5.93 | 12.71 | 41.36 | **18.1** | **9.3** | 2.83 | 44.8 | **82.6** | **87.7** | 34.4 |
| $\lambda$=0.9 | **7.47** | 10.98 | 41.98 | 21.2 | 10.7 | 2.76 | 46.9 | 81.8 | 87.2 | 34.2 |

Accordingly, the gradient of $\mathcal{L}^*_{VCO}$ with respect to $\theta$ is then given by:

$$
\begin{aligned}
\frac{\partial \mathcal{L}^*_{VCO}}{\partial \theta} &= \frac{\partial \mathcal{L}^*_{VCO}}{\partial u} \cdot \frac{\partial u}{\partial \theta} \\
&= -\frac{\partial}{\partial u} \log \sigma(u+c) \cdot \frac{\partial u}{\partial \theta} \\
&= -\big(1 - \sigma(u+c)\big) \cdot \frac{\partial u}{\partial \theta} \\
&= -\sigma\big(-(u+c)\big) \cdot \frac{\partial u}{\partial \theta}.
\end{aligned}
\tag{14}
$$

Similarly, we can derive the gradient of the visual-oriented contrastive objective used in existing works (i.e., Equation 6) as:

$$
\frac{\partial \mathcal{L}_{VCO}}{\partial \theta} = -\sigma(-u)\frac{\partial u}{\partial \theta}.
\tag{15}
$$

By comparing the gradients in Equations 14 and 15, we observe that the constant $c$ in Equations 14 acts as an offset modulating the coefficient of the gradient term $\frac{\partial u}{\partial \theta}$. Intuitively, since $Z(m, x)$ integrates over all possible responses $y$, it inherently captures the global quality of the response space conditioned on $(m, x)$. Thus, $c$ reflects the reference model's global response quality discrepancy between the contexts $(m_w, x)$ and $(m_l, x)$. Specifically, a larger value of $c$ indicates a stronger discrepancy in model behavior between $(m_w, x)$ and $(m_l, x)$, exerting greater influence on gradient adjustment for preference alignment optimization. Conversely, a smaller value of $c$ implies a weaker response quality gap between the multimodal context pair, less impacting the gradient update during training. However, existing methods neglect the two partition functions in visual-oriented contrastive optimization. In essence, these methods assign the same value (i.e., $c = 0$) to all contrastive samples, preventing adaptive weight adjustment for different image contexts. Consequently, the model fails to achieve optimal visual understanding capability, as it tends to either over-attend to images with limited informative signals or under-reason about images containing complex visual clues.

## C  Additional Ablation Studies

In this section, we detailed the process of selecting hyper-parameters for our experiments.

For the hyper-parameters $\beta$, $\eta$ and $\delta$, we adopt the same setup as prior works. Specifically, $\beta$ is directly set to 0.1, while $\eta$ and $\delta$ are chosen to be 1.0 and 0, respectively, in alignment with the configurations used in mDPO and OPA-DPO. For the other hyper-parameters, including the learning rate, $\lambda$ and $\gamma$, we determine their values through a grid search. The search is conducted over the following ranges: the learning rate is evaluated at [5e-5, 5e-6, 5e-7], $\lambda$ is tested across [0.1, 0.3,

Table 5: Comparison of different $\gamma$ on the performance of SymMPO

| SymMPO | HallusionBench | | | Object-HalBench | | MMHal-Bench | | AMBER | | MMStar |
|---|---|---|---|---|---|---|---|---|---|---|
| | qAcc↑ | fAcc↑ | aAcc↑ | Resp.↓ | Ment.↓ | Score↑ | Hall↓ | Acc↑ | F1↑ | Overall↑ |
| $\gamma$=1e-2 | 4.39 | 10.40 | 40.83 | 24.4 | 15.2 | 2.15 | 58.3 | 78.9 | 83.5 | **34.9** |
| $\gamma$=1e-3 | 6.59 | 10.40 | 43.13 | 21.0 | 9.9 | 2.70 | 49.0 | **83.1** | 87.6 | **34.9** |
| $\gamma$=1e-4 | **7.25** | **13.58** | **44.28** | **19.5** | **9.7** | **2.89** | **42.7** | 82.6 | 87.7 | 34.8 |
| $\gamma$=1e-5 | 6.37 | 10.98 | 43.75 | 21.0 | 10.4 | 2.70 | 47.9 | 82.4 | **87.8** | 33.2 |

0.5, 0.7, 0.9], and $\gamma$ is explored within [1e-2, 1e-3, 1e-4, 1e-5]. The results from this grid search are presented in Table 3, Table 4, and Table 5, respectively, with all experiments conducted using LLaVA-1.5-7B. Based on these results, the final hyper-parameter values chosen for our experiments are as follows: a learning rate of 5e-6, $\lambda$=0.5, and $\gamma$=1e-4.

Compared to the learning rate and $\lambda$, the hyper-parameter $\gamma$, which governs preference margin consistency regularization, is notably smaller. This may be attributed to the subtle and sensitive nature of relative relationships between preferences across contrastive images. As a result, smaller $\gamma$ values lead to weaker regularization of preference margin consistency, limiting the effectiveness of SymMPO. Conversely, larger $\gamma$ values impose excessively strong regularization, which can disrupt preference learning. These observations highlight the critical need to carefully balance the regularization strength to achieve optimal performance.

## D  Case Study

In this part, we present the case study on our constructed preference pair data for model training and generated responses for model evaluation.

**On Preference Pair Construction.** As illustrated in Figure 1, each training sample for our symmetric preference optimization comprises two triplets sharing the same prompt and similar but distinct image-response pairs. Empirically, the contrastive images are identified using CLIP similarity scores, while the responses are generated using the caption-anchored response preference pair construction pipeline described in Section 4.1 for corresponding images given the same prompt. Figure 5 illustrates two training examples from our constructed preference dataset, where claims in each response that contradict (and are thus potentially hallucinated relative to) the paired response are highlighted in red. These two examples demonstrate that the effectiveness of using CLIP similarity to obtain contrastive image pairs with subtle visual differences, and confirm that the corresponding responses generated by our preference data construction pipeline indeed exhibit strong linguistic alignment with only minor claim variations. Overall, these examples show the effectiveness of our training preference pair construction strategy.

**On Hallucination Mitigation.** To demonstrate the advantages of SymMPO, we compare the specific responses generated by original LLaVA-1.5-7B and our SymMPO-enhanced version with samples from two widely used benchmarks, MMHal-Bench and Object-HalBench. Both benchmarks feature diverse visual questions closely related to daily life. Figure 6 and Figure 7 present the corresponding results, with hallucinated content highlighted in red. As can be seen, LLaVA-1.5-7B+SymMPO consistently generates more accurate responses than the original LLaVA-1.5-7B across both benchmarks. Even for visual questions requiring detailed responses in Object-HalBench, LLaVA-1.5-7B+SymMPO maintains precise, hallucination-free responses, while LLaVA-1.5-7B fails by producing responses with hallucinations. These results confirm the effectiveness of our SymMPO in enhancing vision-language understanding and mitigating MLLM hallucinations.

**Case 1**  **Prompt:** How do these people know each other?

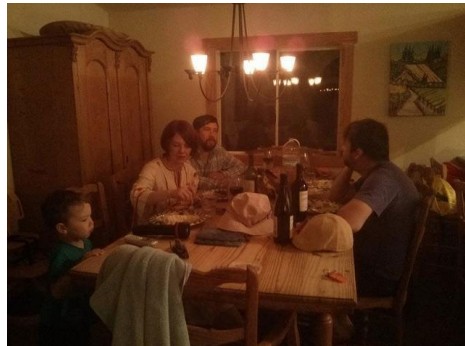

**Original Image**  **Contrastive Image**

**Response 1 (For Original Image):** These people know each other as they are a family, consisting of two men and a woman, who have gathered around a wooden dining table to share a meal and spend quality time together. They are sitting on three different chairs (green, yellow-green, and red), and the table is set with various items such as plates, bowls, knives, spoons, glasses, and condiments/bottles. This indicates that they are enjoying a dinner together, which is a common activity for families to bond and strengthen their relationships.

- - - - - - - - - - - - - - - - - - - - - - - - - - - - - - - - - - - - - - - - - - - - -

**Response 2 (For Contrastive Image):** The group of people in the image, consisting of two men, a woman, and a child, are likely related or close friends, as they are gathered together for a meal. They are sitting around a dining table with two dark-colored bottles. The presence of food and drinks, coupled with the comfortable seating arrangement, indicates that they are part of a social gathering or celebration. It is likely that they are enjoying each other's company and sharing a meal, which is a common way to bond and strengthen relationships.

---

**Case 2**  **Prompt:** What are the main objects or subjects in the image?

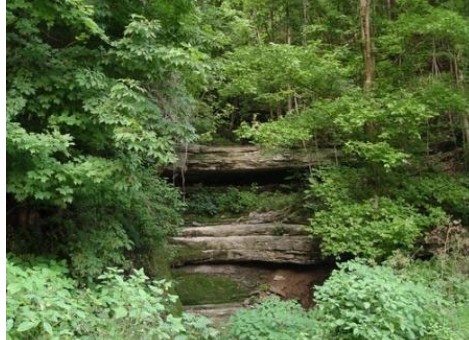 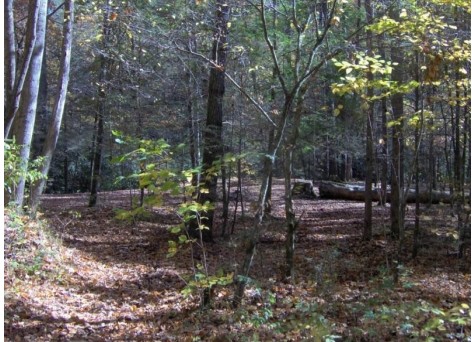

**Original Image**  **Contrastive Image**

**Response 1 (For Original Image):** The main objects or subjects in the image are a dense forest with tall, leafy trees featuring vibrant green foliage, suggesting spring or summer. There is a rocky outcropping composed of layered ledges in light gray to tan hues, with distinct horizontal stratification and partial moss coverage. The ground is covered with grass, moss, and low-lying plants, along with patches of darker soil near the base of the rocks.

- - - - - - - - - - - - - - - - - - - - - - - - - - - - - - - - - - - - - - - - - - - - -

**Response 2 (For Contrastive Image):** The main objects or subjects in the image are trees, fallen leaves, and a decomposed log. The scene features a forest with mostly slender to medium-sized trees, some with light-colored bark (possibly birch) and others with darker bark. The ground is covered with a thick layer of brown and yellow leaves, indicating autumn, while some green foliage remains on the trees. Sparse shrubs and small plants are visible near the base of some trees. A fallen log lies horizontally in the mid-ground, partially decomposed and blending into the natural environment. Soft, diffused sunlight filters through the canopy, creating dappled shadows on the ground, enhancing the picturesque autumn setting.

Figure 5: Two training samples from our built preference dataset used to optimize SymMPO, with hallucinated elements relative to the other response highlighted in red.

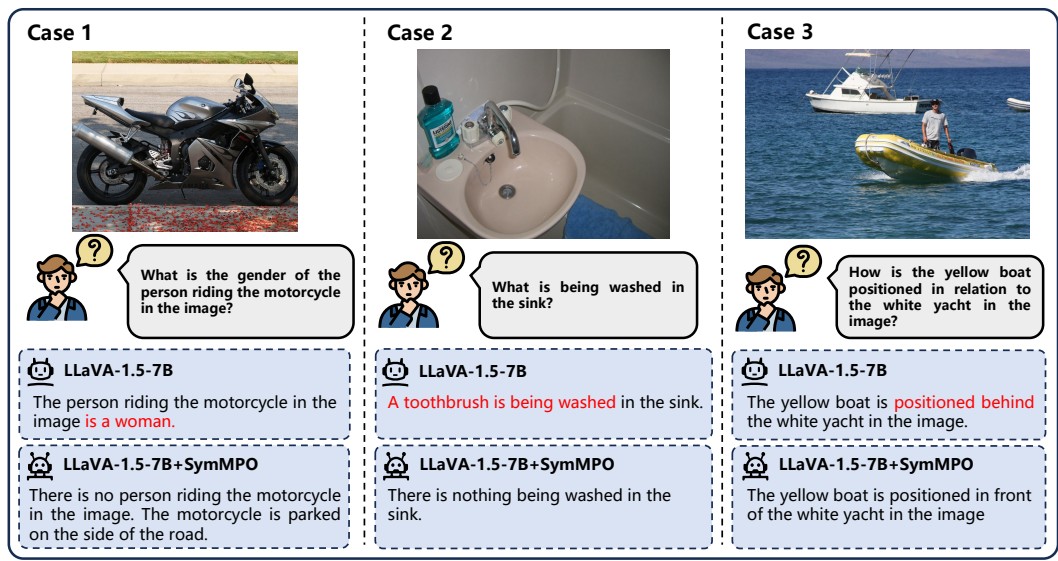

Figure 6: Responses generated by LLaVA-1.5-7B and our SymMPO-enhanced version for data examples from MMHal-Bench, with hallucinated contents highlighted in red.

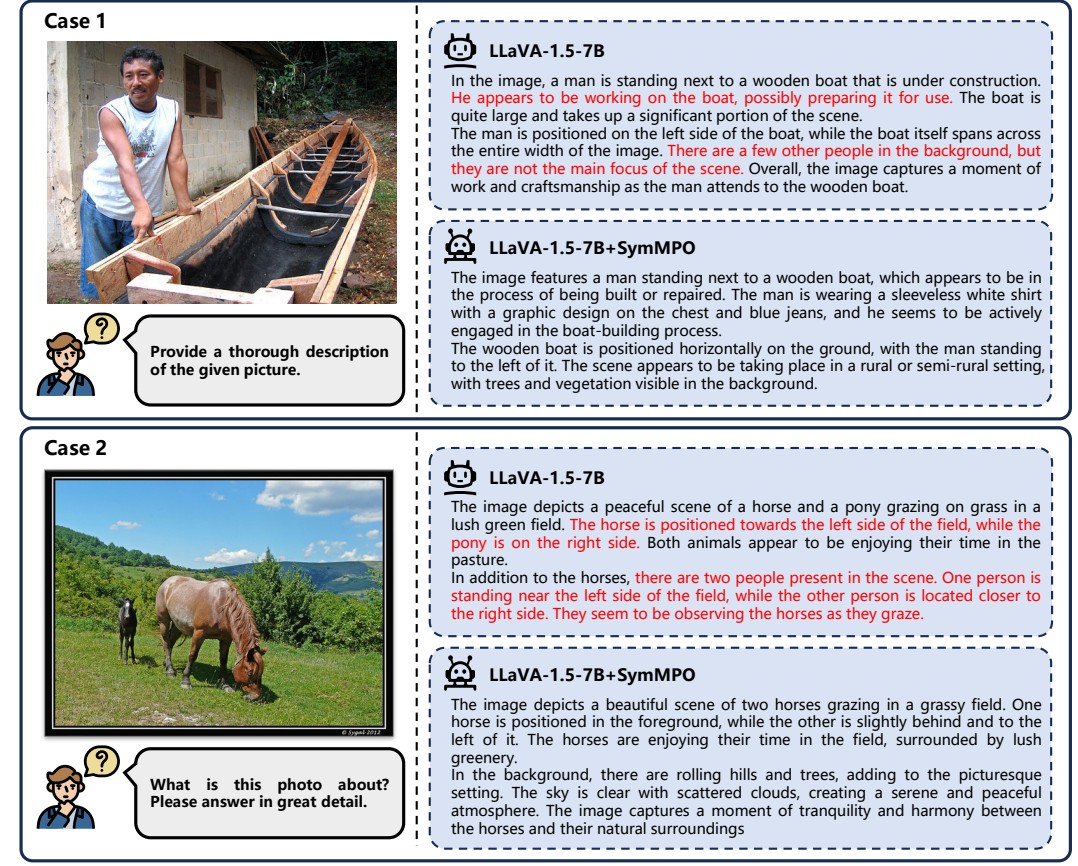

Figure 7: Responses generated by LLaVA-1.5-7B and our SymMPO-enhanced version for data examples from Object-HalBench, with hallucinated contents highlighted in red.

