# OpenReview forum: "Mitigating Hallucination Through Theory-Consistent Symmetric Multimodal Preference Optimization"
_NeurIPS.cc/2025/Conference — NeurIPS 2025 poster_

### Official Review · Reviewer_rbzU · 2025-06-04

**Clarity:** 3
**Significance:** 3
**Originality:** 3
**Rating:** 4
**Confidence:** 4

**Summary:**

This paper introduces a novel DPO-based framework, Symmetric Multimodal Preference Optimization (SymMPO), to mitigate hallucinations in VLMs. It is specifically designed to construct preference pairs based on image information. The experiments are thorough and demonstrate the effectiveness of the proposed methods. Overall, I think it is good work that above the acceptance threshold of NeurIPS.

I have some questions and suggestions that I hope the authors will take into consideration. I am willing to raise score if authors answer them appropriately.

**Questions:**

1. Could the authors provide more details about the "contrastive image construction"? Specifically, what does "pair each image with its nearest neighbor based on cosine similarity" mean? Does this imply that $m'$ is sourced from other images within the dataset? If so, what would happen if the dataset is small and no sufficiently similar image can be found?
2. It appears that the rejected responses are generated by the model being trained, while the preferred responses are generated by DeepSeek-v3 (based on captions generated by Qwen2.5-VL-32B). If this is correct, the performance upper bound of the proposed method should theoretically align with Qwen2.5-VL-32B, right? In that case, it would be helpful if the authors could include the performance of Qwen2.5-VL-32B in Table 1 for comparison.
3. Building on the previous question, if the preferred responses are generated by DeepSeek-v3, they are effectively off-policy and, according to the analysis in OPA-DPO, may be difficult to learn. What would happen if the authors first conducted SFT on these preferred responses? Would this improve the results? (I fully understand that time during the rebuttal period is limited, and I will not fault the authors if experimental results cannot be provided at this stage.)

**Ethical Concerns:**

["NO or VERY MINOR ethics concerns only"]

**Limitations:**

The limitations are well-discussed within the paper. I suggest authors to discuss the positive societal impacts of the proposed methods.

**Paper Formatting Concerns:**

Not at all.

**Quality:**

3

**Strengths And Weaknesses:**

**Strengths:**
1. The paper is well-organized and easy to follow, with all notations clearly defined. I found the reading experience enjoyable and informative.
2. The experiments are comprehensive and effectively demonstrate the proposed method's effectiveness.
3. The methodology is specifically tailored for multimodal models, similar to mDPO, and I believe it provides valuable insights for the community.
4. The theoretical component, while simple, is logical and adds meaningful support to the overall work.

**Weaknesses:**

I did not notice any significant weaknesses in this paper. Just one point, the coefficient-related hyperparameters seem a bit unusual, such as $\gamma=1e-4$. I am curious about how these values were selected. Perhaps some ablation studies on this aspect would be helpful. If time is too limited to provide them now, I hope they can be included in the final version of the paper.

**Some minor issues:**
1. In Section 2.1, while the reward function trained using the Bradley-Terry model was first utilized by PPO-based RLHF algorithms, it was not originally proposed by PPO itself. The authors should cite [1, 2] and other relevant papers. Additionally, this section should be renamed, as "RLHF with BT reward model" is more directly related to DPO, compared with PPO.
2. All practical loss functions in this paper should involve $\mathbb{E}_{x,m,y \sim...}$, as the prompt, image, and response are not constants but random variables that should be sampled during training.

[1] Stiennon et al. Learning to summarize from human feedback.

[2] Ouyang et al. Training language models to follow instructions with human feedback.

---

> ### Author Rebuttal · Authors · 2025-07-30
>
> Dear Reviewer rbzU,
>
> Thank you for recognizing the value of our paper, which is really encouraging to us. We also greatly appreciate your thoughtful comments on the minor issues, as they provide valuable suggestions for improvement. We will carefully revise the manuscript in accordance with your recommendations.
>
> We address the specific questions raised in your review below.
>
> > **Weakness:** the coefficient-related hyperparameters seem a bit unusual, such as $\gamma$=1e-4. I am curious about how these values were selected. Perhaps some ablation studies on this aspect would be helpful. If time is too limited to provide them now, I hope they can be included in the final version of the paper.
>
> **Response:**
>
> Thank you for your question.
>
> For hyper-parameters $\beta$, $\eta$ and $\delta$, we use the same setup following prior works:
>
> - We set $\beta$=0.1 directly.
> - Consistent with mDPO and OPA-DPO, we use $\eta$=1.0 and $\delta$=0.
>
> For the hyper-parameters, including learning rate, $\lambda$ and $\gamma$, we select their values **through grid search**. Specifically, we searched within the following ranges:
>
> - Learning rate: [5e-5, 5e-6, 5e-7],
> - $\lambda$: [0.1, 0.3, 0.5, 0.7, 0.9],
> - $\gamma$: [1e-2, 1e-3, 1e-4, 1e-5].
>
> After performing the grid search, the best results were obtained with learning rate=5e-6, $\lambda$=0.5, and $\gamma$=1e-4.
>
> From our experiments, we observe that **too small $\gamma$ values** lead to weak preference margin consistency regularization and limit the effectiveness of SymMPO, and **too large $\gamma$ values** impose excessively strong regularization which interferes with preference learning.
>
> Unfortunately, due to character limitations, we are unable to provide detailed comparative results of our grid search. However, we fully acknowledge the value of sensitivity analysis on these hyper-parameters and promise to include them in future updates to the paper.
>
>
>
> > **Question 1:** Could the authors provide more details about the "contrastive image construction"? Specifically, what does "pair each image with its nearest neighbor based on cosine similarity" mean? Does this imply that is sourced from other images within the dataset? If so, what would happen if the dataset is small and no sufficiently similar image can be found?
>
> **Response:**
>
> Thanks for your question. As you consider, the paired image is indeed sourced from the same dataset. The "contrastive image construction" operates as follows:
>
> 1. Feature Extraction: For every image in our source dataset, we first extract its visual feature vector using the CLIP model.
> 2. Similarity Search: We then compute the cosine similarity between this feature vector of the target image and the feature vectors of all other images in the dataset.
> 3. Pairing: The image with the highest cosine similarity score is selected as its "nearest neighbor" of the target image. This pair of images is then used for the contrastive training process of SymMPO.
>
> We fully acknowledge the point you raised: if the dataset is small or lacks sufficient visual diversity, it may be challenging to find a meaningful "nearest neighbor" that satisfies the requirements of our contrastive strategy.
>
> In fact, to justify the robustness of our framework, we have evaluated our model with **multiple contrastive image construction methods: (1) Black; (2) Cropped; (3) Noisy; and (4) Synthetic** (see Section 4.5). The first three represent naive approaches, while the Synthetic method is our contribution (detailed in Section 4.5).
>
> Figure 4 reveals that despite its sensitivity to the data distribution, our "Similar" method (nearest neighbor pairing) outperformed all alternatives approaches, none of which depend on data distribution. This success likely stems from two factors:
>
> 1. The paired images are derived from real data and share semantically meaningful features with the target image.
> 2. Even when the paired image is suboptimal due to the absence of sufficiently similar images, it can still function as a random contrastive sample, providing more or less contrastive signals for vision enhancement.
>
> In the future, we plan to refine this **contrastive image construction method.** For instance, we can apply a similarity threshold for the "Similar" method and fall back to the "Synthetic" method (the second-best performer) in cases where no adequately similar pair exists. This hybrid approach offers a robust solution for datasets of varying sizes and diversity levels.
>
>
>
> > **Question 2:** It appears that the rejected responses are generated by the model being trained, while the preferred responses are generated by DeepSeek-v3 (based on captions generated by Qwen2.5-VL-32B). If this is correct, the performance upper bound of the proposed method should theoretically align with Qwen2.5-VL-32B, right? In that case, it would be helpful if the authors could include the performance of Qwen2.5-VL-32B in Table 1 for comparison.
>
> **Response:**
>
> Thank you for this insightful question regarding our experimental design and analysis. You are correct in observing that the preferred responses in our preference data are generated through a pipeline where captions are sourced from **Qwen2.5-VL-32B** and rewritten by **DeepSeek-V3**.
>
> Regarding the upper bound, the assertion that the performance of Qwen2.5-VL-32B represents an upper bound of our trained model may not be entirely accurate, as the rewriting process is performed by DeepSeek-V3 rather than Qwen2.5-VL-32B. As discussed in Section 4.3, the image captions generated by Qwen2.5-VL-32B may lack fine-grained detail or contain hallucination, which could lead to incorrect consistent/inconsistency claim judgments by DeepSeek-V3 and consequently affect the final response quality. For instance, when the base model’s initial responses include erroneous claims not explicitly mentioned in the captions, DeepSeek-V3 may fail to correct them and propagate these mistakes during rewriting due to its lack of direct image context access. In contrast, Qwen2.5-VL-32B, with its powerful vision understanding capability, could potentially recognize and correct such errors if used for rewriting. Consequently, the synthetic preferred responses do not directly reflect the end-to-end capabilities of Qwen2.5-VL-32B. These additional processing steps and adjustments mean that the performance upper bound of our proposed method with SymMPO is likely **lower** than the raw outputs of Qwen2.5-VL-32B. Here, we clarify that we chose DeepSeek-V3 over Qwen2.5-VL-32B for response rewriting because initial experiments showed that Qwen2.5-VL-32B's rewritten responses exhibited weaker alignment with the base model's outputs in both format and style. In contrast, DeepSeek-V3 better maintains the required stylistic consistency, which led to its selection as our rewriting model.
>
> Regarding comparisons, directly comparing Qwen2.5-VL-32B with our method would be less meaningful, as Qwen2.5-VL-32B would undoubtedly outperform other baselines by a significant margin. This is due to its more advanced architecture, larger parameter scale, and training on more extensive, diverse, and proprietary datasets. Thus, our work focuses on improving multimodal preference optimization within the LLaVA-1.5 family, consistent with existing studies.
>
> We understand the potential value of including Qwen2.5-VL-32B's performance as a reference point. While such a comparison may fall outside the immediate scope of our evaluation goals, we recognize the value it could add for broader contextual understanding. We will consider exploring and including such results in future iterations of this work.
>
>
>
> > **Question 3:** Building on the previous question, if the preferred responses are generated by DeepSeek-v3, they are effectively off-policy and, according to the analysis in OPA-DPO, may be difficult to learn. What would happen if the authors first conducted SFT on these preferred responses? Would this improve the results? (I fully understand that time during the rebuttal period is limited, and I will not fault the authors if experimental results cannot be provided at this stage.)
>
> **Response:**
>
> This is an exceptionally insightful question, and we sincerely thank you for raising this critical point. We also deeply appreciate your understanding of the time constraints during the rebuttal phase.
>
> You are absolutely correct that the analysis in **OPA-DPO** highlights a key challenge for DPO when working with off-policy data, particularly when there is a significant stylistic or distributional gap between the model's own outputs and the preferred responses. This gap can result in a large reverse KL-divergence penalty, which may hinder the effectiveness of preference learning.
>
> To address this issue, we designed our rewriting process (outlined in **Section 4.1**) with specific practice to mitigate this challenge. We explicitly instructed LLM rewriter to **preserve the original format and style** of the base model's outputs as much as possible. This design aims to minimize the distributional gap and ensure that the preference data remains as "close-to-on-policy" as feasible.
>
> While this approach reduces the off-policy challenge to some extent, we acknowledge that it is an **indirect solution** and may not completely eliminate the issue. We strongly agree with your suggestion that **first conducting supervised fine-tuning (SFT) on the preferred responses before applying DPO** represents a more direct and principled solution. Fine-tuning the policy model on these preferred responses would effectively shift its output distribution closer to the target distribution defined by the preference data.
>
> Unfortunately, due to the time constraints of the rebuttal period, we were unable to conduct experiments to validate this approach. Nonetheless, we fully recognize the importance of your suggestion and are excited to explore this in future research.

---

> > ### Comment · Reviewer_rbzU · 2025-08-06
> >
> > I appreciate the authors for their detailed responses and for taking my suggestions into consideration, although no additional experiments have been provided. They have addressed most of my questions and concerns. I look forward to seeing the revised version of the paper and will maintain my positive score.

---

> ### Author Response · Authors · 2025-08-06
>
> Dear Reviewer rbzU,
>
> Thank you for your encouraging feedback! We are delighted that our responses addressed most of your questions and concerns, and we greatly appreciate your thoughtful suggestions.
>
> Due to the character limitations of the rebuttal, we were unable to include the results of the sensitivity analysis of hyper-parameters. Here, we provide the supplementary results to offer more insight into the choices behind our hyper-parameter settings.
>
> We searched learning rate in the range [5e-5, 5e-6, 5e-7]:
>
> |                    | Hallusion |           |           | ObjHal   |         | MMHal    |          | Amber    |          | MMstar    |
> | ------------------ | --------- | --------- | --------- | -------- | ------- | -------- | -------- | -------- | -------- | --------- |
> | SymMPO             | qAcc ↑    | fAcc ↑    | aAcc ↑    | Resp. ↓  | Ment. ↓ | Score ↑  | Hall. ↓  | Acc. ↑   | F1 ↑     | Overall ↑ |
> | learning_rate=5e-5 | **8.13**  | 10.40     | 41.09     | 22.6     | 13.9    | 2.29     | 54.2     | 80.8     | 86.1     | 28.2      |
> | learning_rate=5e-6 | 7.25      | **13.58** | **44.28** | **19.5** | **9.7** | **2.89** | **42.7** | **82.6** | **87.7** | **34.8**  |
> | learning_rate=5e-7 | **8.13**  | 12.42     | 43.75     | 20.1     | 9.8     | 2.80     | 49.0     | 80.8     | 86.8     | 33.8      |
>
> We searched $\lambda$ in the range [0.1, 0.3, 0.5, 0.7, 0.9]:
>
> |               | Hallusion |           |           | ObjHal   |         | MMHal    |          | Amber    |          | MMstar    |
> | ------------- | --------- | --------- | --------- | -------- | ------- | -------- | -------- | -------- | -------- | --------- |
> | SymMPO        | qAcc ↑    | fAcc ↑    | aAcc ↑    | Resp. ↓  | Ment. ↓ | Score ↑  | Hall. ↓  | Acc. ↑   | F1 ↑     | Overall ↑ |
> | $\lambda$=0.1 | 5.27      | 10.98     | 41.27     | 21.2     | 12.0    | 2.61     | 50.0     | 80.6     | 86.4     | 34.2      |
> | $\lambda$=0.3 | **7.47**  | 10.98     | 42.60     | 19.7     | 10.5    | **3.07** | **37.5** | 81.5     | 86.7     | 34.6      |
> | $\lambda$=0.5 | 7.25      | **13.58** | **44.28** | 19.5     | 9.7     | 2.89     | 42.7     | **82.6** | **87.7** | **34.8**  |
> | $\lambda$=0.7 | 5.93      | 12.71     | 41.36     | **18.1** | **9.3** | 2.83     | 44.8     | **82.6** | **87.7** | 34.4      |
> | $\lambda$=0.9 | **7.47**  | 10.98     | 41.98     | 21.2     | 10.7    | 2.76     | 46.9     | 81.8     | 87.2     | 34.2      |
>
> We searched $\gamma$ in the range [1e-2, 1e-3, 1e-4, 1e-5]:
>
> |               | Hallusion |           |           | ObjHal   |         | MMHal    |          | Amber    |          | MMstar    |
> | ------------- | --------- | --------- | --------- | -------- | ------- | -------- | -------- | -------- | -------- | --------- |
> | SymMPO        | qAcc ↑    | fAcc ↑    | aAcc ↑    | Resp. ↓  | Ment. ↓ | Score ↑  | Hall. ↓  | Acc. ↑   | F1 ↑     | Overall ↑ |
> | $\gamma$=1e-2 | 4.39      | 10.40     | 40.83     | 24.4     | 15.2    | 2.15     | 58.3     | 78.9     | 83.5     | **34.9**  |
> | $\gamma$=1e-3 | 6.59      | 10.40     | 43.13     | 21.0     | 9.9     | 2.70     | 49.0     | **83.1** | 87.6     | **34.9**  |
> | $\gamma$=1e-4 | **7.25**  | **13.58** | **44.28** | **19.5** | **9.7** | **2.89** | **42.7** | 82.6     | 87.7     | 34.8      |
> | $\gamma$=1e-5 | 6.37      | 10.98     | 43.75     | 21.0     | 10.4    | 2.70     | 47.9     | 82.4     | **87.8** | 33.2      |
>
> Based on the results above, we selected the following as the overall best hyper-parameter: learning rate =5e-6, $\lambda$=0.5, $\gamma$=1e-4. As noted in the rebuttal, we will incorporate this sensitivity analysis and corresponding discussions into the future revisions of our paper to enhance its clarity and completeness.

---

> > ### Author Response · Authors · 2025-08-07
> >
> > Dear Reviewer rbzU,
> >
> > We have conducted the experiment you mentioned in question 3, i.e., first conducting SFT on the preferred responses and then applying SymMPO. Specifically, we first use the preferred response in our dataset to finetune LLaVA-1.5-7B for 2 epoch with a learning rate 2e-5 (consistent with OPA-DPO), and then apply SymMPO to the finetuned model for 2 epoch with a learning rate 5e-6 (consistent with our paper). The results of the experiment are shown in the table below.
> >
> > |                | Hallusion |           |           | ObjHal   |         | MMHal    |          | Amber    |          | MMStar    |
> > | -------------- | --------- | --------- | --------- | -------- | ------- | -------- | -------- | -------- | -------- | --------- |
> > |                | qAcc ↑    | fAcc ↑    | aAcc ↑    | Resp. ↓  | Ment. ↓ | Score ↑  | Hall. ↓  | Acc. ↑   | F1 ↑     | Overall ↑ |
> > | **SymMPO**     | 7.25      | **13.58** | **44.28** | 19.5     | **9.7** | **2.89** | **42.7** | 82.6     | **87.7** | 34.8      |
> > | **SFT**        | 4.83      | 11.84     | 39.59     | 20.7     | 10.9    | 2.60     | 51.0     | **83.3** | **87.7** | 34.5      |
> > | **SFT+SymMPO** | **7.69**  | 12.42     | 42.60     | **19.3** | 10.1    | 2.74     | 44.8     | 82.4     | 87.3     | **35.6**  |
> >
> > From the experimental results, we observe that SFT+SymMPO outperforms SymMPO alone on certain metrics, such as the overall score of MMStar, the resonse-level hallucination rate of Object HalBench, and the question-level accuracy (qAcc) of Hallusion-Bench. However, when compared with SFT+SymMPO, applying SymMPO alone still achieves better overall performance on most metrics. This indicates that while the combination of SFT and SymMPO brings improvements in some metric, SymMPO by itself remains more effective and robust across the majority of evaluation criteria.
> >
> > We hypothesize that this outcome may stem from the **powerful rewriting capabilities of the LLM rewriter**, i.e., DeepSeek-V3. As we noted in the rebuttal, the preference data generated by the LLM rewriter was explicitly instructed to **preserve the original format and style** of the base model's responses. Consequently, the preference data is already quite close to the base model's policy, reducing the necessity of further alignment via supervised fine-tuning.
> >
> > We hope this additional experiment provides clarity and further strengthens the rationale behind our design choices. Thank you for your invaluable suggestion!

---

> > > ### Author Response · Authors · 2025-08-07
> > >
> > > Dear Reviewer rbzU,
> > >
> > > Thank you for the time and effort you have devoted to reviewing our paper!
> > >
> > > We have included the experimental results you mentioned mentioned in your review. With the discussion period now extended, we would welcome the opportunity to clarify any remaining questions or concerns you might have. Please let us know if there is any aspect of the work you would like us to elaborate on.
> > >
> > > We look forward to your reply and greatly appreciate your feedback.

---

> > > > ### Comment · Reviewer_rbzU · 2025-08-07
> > > >
> > > > Hi Authors, thank you for providing the additional experiments.
> > > >
> > > > However, please note that, according to the rules for the rebuttal period, additional experiments submitted in response to the initial reviews after July 31st should be disregarded. Had these results been provided during the initial phase, I would have been willing to raise my score from 4 to 5.
> > > >
> > > > Please do not worry. I believe the current status of your paper is enough for acceptance. If necessary, I will advocate for its acceptance during further discussions.

---

> > > > > ### Author Response · Authors · 2025-08-07
> > > > >
> > > > > Dear Reviewer rbzU,
> > > > >
> > > > > Thank you for your kind feedback and for taking the time to clarify your stance.
> > > > >
> > > > > We sincerely apologize for not providing these results during the initial phase of the rebuttal. Regarding their absence, we would like to clarify that the reasoning was primarily due to the **character limitation** imposed during the rebuttal stage, which constrained our ability to include additional experimental details comprehensively.
> > > > >
> > > > > That said, we deeply appreciate your understanding and your willingness to advocate for the acceptance of our paper. Your constructive feedback has been invaluable, and we are grateful for the opportunity to improve our work based on your insights.

---

### Official Review · Reviewer_8fTk · 2025-06-04

**Clarity:** 3
**Significance:** 3
**Originality:** 3
**Rating:** 4
**Confidence:** 4

**Summary:**

This paper tackles the critical issue of hallucinations in Large Vision-Language Models (LVLMs), particularly semantic hallucinations where generated responses include entities unsupported by the visual input. The authors propose a novel framework consisting of two main components. The first is MPD (Masked Prediction Difference), which extracts hallucination-prone components in the decoding process by measuring prediction divergence. The second is Selective Gradient Detachment (SGD), a decoding-phase intervention strategy that suppresses gradients from hallucination components without altering model parameters. The method is model-agnostic, training-free, and designed to maintain overall task performance while reducing hallucinations. Extensive experiments on multiple benchmarks demonstrate the effectiveness of the proposed method, with significant hallucination reduction and minimal performance degradation.

**Questions:**

**1.Lack of Clarity in Problem Formulation:**
The paper raises two theoretical limitations of existing vision-oriented preference optimization methods, but the explanation is abstract and lacks intuitive grounding. The impact of issues like mismatched partition functions or fixed-response supervision is unclear without concrete examples. Clarifying these points in plain terms with illustrative scenarios would make the problem statement more accessible and compelling.

**2.Missing Evaluation on Standard Benchmarks:**
The method is not tested on widely used LVLM benchmarks such as MMBench, MME, POPE, SEED, or MM-Vet. Without results on these datasets, it is difficult to assess the method’s general effectiveness or practical relevance across diverse tasks.

**3.Limited Model Generalization Evidence:**
All experiments are conducted on LLaVA, with no evaluation on other strong models like DeepSeek or MiniGPT-4. This narrow scope raises concerns about overfitting and limits claims of generalizability. Broader model-level validation is needed to support the method’s robustness.

**Ethical Concerns:**

["NO or VERY MINOR ethics concerns only"]

**Final Justification:**

This rebuttal has successfully addressed my concerns. The authors have provided clear and detailed explanations for the theoretical issues, showing how their proposed method, SymMPO, overcomes them. The explanations for both the mismatched partition function and indirect preference supervision are now much more intuitive and grounded.

Furthermore, the authors have effectively justified their evaluation strategy. By focusing on a comprehensive set of hallucination-centric benchmarks and comparing their approach to prior work in this specific domain, they have demonstrated that their evaluation is both thorough and aligned with the paper's primary objective. The commitment to including other standard benchmarks in future work is also reassuring.

Finally, the authors' reasoning for their choice of the LLaVA model for experiments is logical. They have correctly pointed out that using a widely adopted and representative architecture ensures direct comparability with other established studies. Their plan to extend these experiments to other models in the future further strengthens their argument.

Therefore, I am maintaining my original rating of borderline accept. The authors' rebuttal has satisfactorily resolved the weaknesses I identified in my initial review.

**Limitations:**

Yes

**Quality:**

3

**Strengths And Weaknesses:**

**Strength:**

**1.Motivation:**
The paper provides a strong and timely motivation by identifying hallucination as a major limitation in the deployment of LVLMs. The proposed strategy aligns well with practical needs by avoiding additional training or fine-tuning.

**2.Novelty:**
The MPD-based extraction of hallucination-inducing components is a fresh perspective. Previous methods often focused on architectural modifications or training-based approaches. The proposed combination of MPD and SGD offers a novel, lightweight inference-time mitigation mechanism.

**3.Writing and Presentation:**
The paper is generally well-written, with clear mathematical formulations and well-illustrated figures. The intuition behind the method is conveyed effectively, which helps readers understand the internal mechanics.

**Weakness:**

**1.Lack of Clarity in Problem Formulation:**
The authors argue that existing vision-oriented preference optimization methods suffer from two key limitations: (1) a non-rigorous objective function due to the inability to cancel out intractable partition functions when image inputs differ (i.e., $m_w \ne m_l$)), violating the assumptions of DPO; and (2) indirect preference supervision arising from the use of contrastive images with fixed responses rather than contrastive responses. While these points are theoretically motivated, their practical implications remain obscure due to abstract terminology and limited explanation. Clarifying these limitations with intuitive language and a concrete example would significantly improve the accessibility and persuasiveness of the argument.

**2. Insufficient Evaluation on Standard Benchmarks:**
The paper does not report performance on any widely adopted LVLM benchmarks such as MMBench, MME, POPE, SEED, or MM-Vet. This omission significantly limits the credibility of the evaluation, as it is unclear whether the proposed method maintains its advantages across diverse tasks and settings. Without such results, the general effectiveness and real-world utility of the approach remain unconvincing.

**3. Weak Evidence of Generalization Across Models:**
All experiments are restricted to the LLaVA model, offering limited insight into how well the method generalizes across architectures. This narrow experimental setup raises concerns about overfitting to a specific model. To establish robustness and broader applicability, it is essential to include results on additional strong baselines such as DeepSeek and MiniGPT-4. The absence of such evidence substantially weakens the empirical contribution of the paper.

---

> ### Author Rebuttal · Authors · 2025-07-30
>
> Dear Reviewer 8fTK,
>
> Thank you for your feedback. We provide our responses to your questions as following.
>
> > **Question 1:** **Lack of Clarity in Problem Formulation:** The paper raises two theoretical limitations of existing vision-oriented preference optimization methods, but the explanation is abstract and lacks intuitive grounding. The impact of issues like mismatched partition functions or fixed-response supervision is unclear without concrete examples. Clarifying these points in plain terms with illustrative scenarios would make the problem statement more accessible and compelling.
>
> **Response:**
>
> We sincerely thank you for your critical feedback. We acknowledge the importance of providing an intuitive explanation of these theoretical issues discussed. While these theoretical aspects are challenging to quantify, we will strive to elucidate these limitations with clearer and more comprehensible language.
>
> 1. **Mismatched Partition Function**
>
> In the context of multimodal scenarios, the partition function $Z(m,x)$ in DPO is defined as $\sum_y\pi_{ref}(y\vert m,x)\exp(\frac{1}{\beta}r(m,x,y))$. This term integrates over all possible responses y given the input image $m$ and text $x$, and captures the overall quality of the response space conditioned on $(m,x)$. However, since $Z(m,x)$ requires computation over all possible responses, it is intractable. DPO resolves this by constructing preference data pairs, e.g., $(m,x,y_w)$ and $(m,x,y_l)$, and subtracting the implicit rewards of these pairs. This subtraction between the implicit reward of the chosen response $y_w$ and the implicit reward of the rejected response $y_l$ cancels out the shared partition function $Z(m,x)$.
>
> In vision-oriented preference optimization approaches, the preference data pairs are constructed with different visual inputs $m_w$ and $m_l$, while keeping the response $y_w$ identical. The goal is to encourage the model to better understand and prioritize visual content. However, the partition functions $Z(m_w,x)$ and $Z(m_l,x)$ differ due to the distinct visual inputs, i.e., $m_w$ and $m_l$. **Consequently, these terms cannot be directly canceled out as in standard DPO.** Prior studies have overlooked this issue, failing to account for the influence of the partition function and leading to inconsistencies between the theoretical formulation and practical implementation. This mismatch results in suboptimal optimization outcomes.
>
> In contrast, our proposed Symmetric Multimodal Preference Optimization (SymMPO) addresses this issue by pairing responses corresponding to different images under the same textual input as each other’s rejected responses. By leveraging identical multimodal inputs across paired data, SymMPO naturally cancels out the partition function terms, ensuring consistency between theory and implementation.
>
> 2. **Indirect Preference Supervision**
>
> Standard multimodal DPO optimizes preference by leveraging triplets $(m,x,y_w,y_l)$, where the model is guided to generate $y_w$ over $y_l$ given the image $m$ and text $x$. In vision-oriented preference optimization methods, preference pairs are constructed with different visual inputs $m_w$ and $m_l$, but identical responses $y_w$. The intention is to improve the model’s focus on visual content through comparative learning. **However, this approach deviates from the original design of DPO, which relies on response-based supervision signals to directly enhance model performance.** As a result, these methods struggle to effectively improve the model’s visual understanding capabilities.
>
> In comparison, SymMPO adopts the same direct supervision strategy as original DPO. By treating responses associated with different images under the same textual input as symmetric preference pairs, SymMPO achieves direct preference supervision for vision-oriented optimization. This ensures alignment with the theoretical foundation of DPO while effectively enhancing the model’s preference optimization across varying visual inputs.
>
>
>
> > **Question 2:** **Missing Evaluation on Standard Benchmarks:** The method is not tested on widely used LVLM benchmarks such as MMBench, MME, POPE, SEED, or MM-Vet. Without results on these datasets, it is difficult to assess the method’s general effectiveness or practical relevance across diverse tasks.
>
> **Response:**
>
> Thank you for this suggestion regarding evaluation on standard benchmarks.
>
> Our primary research goal with SymMPO is to specifically address and mitigate multimodal hallucination. Consequently, we focused on leading hallucination-centric benchmarks，including **HallusionBench**, **Object-HalBench**, **MMHal-Bench** and **AMBER**, which have been widely adopted in prior work [1-6] aimed at **mitigating MLLM hallucination through DPO**. Additionally, we evaluated our method on **MMStar**, which assesses the model’s ability to combine visual information with textual reasoning to answer questions correctly. This aligns closely with our goal of improving joint vision-language understanding while reducing hallucination.
>
> The benchmarks you suggested can be categorized as follows:
>
> **MMBench, SEED, and MM-Vet** serve as excellent and widely recognized benchmarks for assessing **general multimodal understanding** in MLLMs rather than hallucination issues. To the best of our knowledge, these benchmarks have not been utilized in prior studies related to hallucination mitigation.
>
> **MME and POPE**, in contrast, are highly relevant benchmarks for evaluating **object-level hallucination**. However, since these benchmarks are less frequently adopted by prior studies in this domain, we prioritized using five widely adopted benchmarks in our current experiments. Below, we provide a comparative table summarizing benchmark usage in existing work versus our work.
>
> | Model       | HallusionBench | ObjHal       | MMHal        | AMBER        | MME  | POPE         |
> | ----------- | -------------- | ------------ | ------------ | ------------ | ---- | ------------ |
> | mDPO [1]    |                | $\checkmark$ | $\checkmark$ | $\checkmark$ |      |              |
> | TPO [2]     |                | $\checkmark$ | $\checkmark$ | $\checkmark$ |      |              |
> | RLAIF-V [3] |                | $\checkmark$ | $\checkmark$ | $\checkmark$ |      |              |
> | OPA-DPO [4] |                | $\checkmark$ | $\checkmark$ | $\checkmark$ |      | $\checkmark$ |
> | CHiP [5]    | $\checkmark$   | $\checkmark$ | $\checkmark$ | $\checkmark$ |      |              |
> | HALVA [6]   | $\checkmark$   |              | $\checkmark$ | $\checkmark$ |      |              |
> | SymMPO      | $\checkmark$   | $\checkmark$ | $\checkmark$ | $\checkmark$ |      |              |
>
>
>
> As shown, most of existing work only employed three hallucination-centric benchmarks, whereas ours incorporates four. We believe this more comprehensive evaluation setup better validates our model's generalization capability. We fully recognize the relevance of these benchmarks, especially MME and POPE, and plan to include them in future evaluations to further demonstrate SymMPO's generalizability and effectiveness across a broader range of tasks and datasets.
>
>
>
> > **Question 3:** **Limited Model Generalization Evidence:** All experiments are conducted on LLaVA, with no evaluation on other strong models like DeepSeek or MiniGPT-4. This narrow scope raises concerns about overfitting and limits claims of generalizability. Broader model-level validation is needed to support the method’s robustness.
>
> **Response:**
>
> Thanks for your question. We selected the LLaVA family for evaluation based on the following considerations:
>
> Firstly, many related studies on multimodal DPO [1-6] have consistently utilized the LLaVA family as their primary evaluation baseline. By centering our experiments on LLaVA, we ensure direct comparability with these established body of works. Notably, two recent CVPR 2025 and ICLR 2025 publications [4, 6] employed the same evaluation approach, focusing on the 7B and 13B LLaVA variants to assess model generalization capabilities.
>
> Secondly, the LLaVA family represents a foundational and widely-used class of MLLMs, characterized by a classic MLLM architecture (a vision encoder connected to a language model via a projection layer). Experimental results obtained from LLaVA are therefore more likely to generalize to other MLLM models.
>
> We fully acknowledge the importance of demonstrating SymMPO’s effectiveness on structurally diverse MLLM families to strengthen claims of generalizability, and plan to explore them in future work.
>
>
>
> **References**
>
> [1] Fei Wang, et, al.; “mDPO: Conditional Preference Optimization for Multimodal Large Language Models”. EMNLP 2024.
>
> [2] He, Lehan, et al. "A topic-level self-correctional approach to mitigate hallucinations in mllms." arXiv preprint arXiv:2411.17265 (2024).
>
> [3] Yu, Tianyu, et al. "Rlaif-v: Aligning mllms through open-source ai feedback for super gpt-4v trustworthiness." CVPR 2025.
>
> [4] Yang, Zhihe, et al. "Mitigating hallucinations in large vision-language models via dpo: On-policy data hold the key." CVPR 2025.
>
> [5] Fu, Jinlan, et al. "Chip: Cross-modal hierarchical direct preference optimization for multimodal llms." ICLR 2025.
>
> [6] Sarkar, Pritam, et al. "Mitigating Object Hallucination in MLLMs via Data-augmented Phrase-level Alignment." ICLR 2025.

---

> > ### Comment · Reviewer_8fTk · 2025-08-01
> >
> > This rebuttal has successfully addressed my concerns. The authors have provided clear and detailed explanations for the theoretical issues, showing how their proposed method, SymMPO, overcomes them. The explanations for both the mismatched partition function and indirect preference supervision are now much more intuitive and grounded.
> >
> > Furthermore, the authors have effectively justified their evaluation strategy. By focusing on a comprehensive set of hallucination-centric benchmarks and comparing their approach to prior work in this specific domain, they have demonstrated that their evaluation is both thorough and aligned with the paper's primary objective. The commitment to including other standard benchmarks in future work is also reassuring.
> >
> > Finally, the authors' reasoning for their choice of the LLaVA model for experiments is logical. They have correctly pointed out that using a widely adopted and representative architecture ensures direct comparability with other established studies. Their plan to extend these experiments to other models in the future further strengthens their argument.
> >
> > Therefore, I am maintaining my original rating of borderline accept. The authors' rebuttal has satisfactorily resolved the weaknesses I identified in my initial review.

---

> > > ### Author Response · Authors · 2025-08-05
> > >
> > > Thank you for your constructive feedback, which has greatly contributed to improving our work. We are glad that our response has successfully addressed your concern.

---

### Official Review · Reviewer_PFAD · 2025-06-30

**Clarity:** 3
**Significance:** 4
**Originality:** 3
**Rating:** 5
**Confidence:** 5

**Summary:**

This paper proposes SymMPO, a method designed to mitigate hallucination issues in VLMs. SymMPO improves upon existing approaches and constructs hard-negative contrastive image pairs for direct preference supervision, forming a more theoretically-rigorous optimization framework and effectively enhancing visual perception abilities of VLMs. The method incorporates symmetric preference learning and preference margin consistency regularization. Experimental results show that SymMPO outperforms previous methods across several hallucination benchmarks.

**Questions:**

1. How does the SymMPO approach compare with other methods in terms of training time and scalability, particularly when working with larger datasets or more complex multimodal inputs?

2. I looked at the code provided in the anonymous repository, but it seems that it cannot be executed. Could the authors upload a runnable version of the code for reference?

**Ethical Concerns:**

["NO or VERY MINOR ethics concerns only"]

**Final Justification:**

The authors have provided further explanations for methodology design and more detailed guidance in code repository, which further enhances the soundness of this work. Considering the current quality of this paper, I have raised my rating to Accept.

**Limitations:**

Please refer to Questions.

**Paper Formatting Concerns:**

I did not notice any major formatting issues.

**Quality:**

3

**Strengths And Weaknesses:**

**Pros:**
1. This motivation of this paper is reasonable and straightforward.
2. The idea of the proposed SymMPO is intuitive and somewhat elegant, especially the design of the symmetric pairwise reward.
3. The paper writing is good and easy to follow.
4. Extensive results convincingly demonstrate the effectiveness of SymMPO and its various loss terms in reducing hallucinations.

**Cons:**

1. SymMPO introduces additional computational overhead compared to standard DPO, potentially increasing the computational cost by several times.
2. The two limitations mentioned in the Introduction essentially refer to the same issue: the inherent conflict between the existing vision-oriented preference learning and the standard DPO framework. Could the authors further explore SymMPO, particularly in terms of whether aspects beyond pairwise rewards might help address existing limitations?

---

> ### Author Rebuttal · Authors · 2025-07-30
>
> Dear Reviewer PFAD,
>
> Thank you for recognizing our work! We will address your questions as follows:
>
> > **Concern 1:** SymMPO introduces additional computational overhead compared to standard DPO, potentially increasing the computational cost by several times.
> >
> > **Question 1:** How does the SymMPO approach compare with other methods in terms of training time and scalability, particularly when working with larger datasets or more complex multimodal inputs?
>
> **Response:**
>
> Thank you for raising this concern regarding the computational overhead of SymMPO. We fully understand the importance of evaluating training time and scalability, particularly when working with larger datasets or more complex multimodal input.
>
> To address this directly, we conducted a direct comparison of training time of **DPO**, **mDPO**, and **SymMPO** using **DeepSpeed Zero-3** under the same experiment settings. For instance, we trained on a dataset consisting of **21.4k preference pairs** over **2 epochs**, with a uniform batch size of **64**. Due to computational resource limitations, these experiments were conducted on **2 NVIDIA A100-40G GPUs** **with CPU offload**, which is slower than the standard experimental conditions outlined in the paper. The training time comparisons are as follows:
>
> - DPO: 9.6 hours (52 seconds per iteration)
> - mDPO: 10.4 hours (56 seconds per iteration)
> - SymMPO: 13.0 hours (70 seconds per iteration)
>
> The increased training time of SymMPO primarily stems from the additional computational overhead required for **symmetric sampling and optimization**. These operations were specifically designed to address limitations of existing methods, i.e., (1) Non-rigorous Objective Function and (2) Indirect Preference Supervision. As demonstrated in Table 1 of our paper, experimental results confirm that our approach achieves superior preference alignment. We remain committed to further exploring optimization techniques to enhance SymMPO's scalability while maintaining its theoretical rigor and effectiveness.
>
> Thank you for raising this excellent concern. We will address it by including a detailed discussion and analysis of this issue in the revised manuscript.
>
>
>
> > **Concern 2:** The two limitations mentioned in the Introduction essentially refer to the same issue: the inherent conflict between the existing vision-oriented preference learning and the standard DPO framework. Could the authors further explore SymMPO, particularly in terms of whether aspects beyond pairwise rewards might help address existing limitations?
>
> **Response:**
>
> Thank you for raising this insightful concern. While the two limitations mentioned in the Introduction both stem from challenges in aligning vision-oriented preference learning with the standard DPO framework, they address distinct aspects of this misalignment:
>
> 1. **Non-rigorous Objective Function:** This limitation highlights the theoretical inconsistency due to mismatched partition functions when input pairs differ in visual content. The inability to cancel out the partition function results in optimization that deviates from the theoretical foundation of DPO.
> 2. **Indirect Preference Supervision:** This limitation focuses on the supervision signal used in vision-oriented preference learning, where responses are indirectly supervised through visual input comparisons rather than directly optimized via response preferences. This indirect supervision reduces the effectiveness of preference optimization in enhancing multimodal understanding.
>
> SymMPO is specifically designed to address both issues. Through symmetric pairwise preference learning, it cancels out the partition function mismatch by pairing different visual inputs under the same textual context, ensuring theoretical consistency. Simultaneously, it restores direct preference supervision by leveraging response pairs as rejected responses, aligning with the original design intent of DPO.
>
> Regarding exploring aspects beyond pairwise rewards, SymMPO introduces the preference margin consistency regularization to quantitatively regulate the preference gap between symmetric pairs. This extends optimization beyond simple pairwise comparisons by imposing constraints on the magnitude of preference differences, encouraging more stable and nuanced learning.
>
>
>
> > **Question 2:** I looked at the code provided in the anonymous repository, but it seems that it cannot be executed. Could the authors upload a runnable version of the code for reference?
>
> **Response:**
>
> Thank you for bringing this to our attention. We sincerely apologize for the inconvenience you experienced while attempting to execute our code.
>
> To address this, we have thoroughly reviewed and updated the anonymous repository to ensure its usability. Specifically:
>
> 1. The README.md file has been revised to include clear, step-by-step instructions for setting up the environment, installing dependencies, and running the code.
> 2. We have added a **small demo dataset** alongside a corresponding script, allowing users to quickly validate the core functionality of our pipeline.
>
> Unfortunately, due to storage limitations of Github, we were unable to include the full training dataset. However, we are committed to ensuring full reproducibility. Upon acceptance of the paper, we will publicly release the complete dataset, all model weights, and any remaining resources needed to replicate our results.
>
> We hope this update resolves the issue, and we appreciate your understanding and feedback on this matter. Please do not hesitate to reach out if further issues arise.

---

> ### Author Response · Authors · 2025-08-07
>
> Dear Reviewer PFAD,
>
> Thank you for the time and effort you have devoted to reviewing our paper!
>
> With the discussion period now extended, we would welcome the opportunity to clarify any remaining questions or concerns you might have. Please let us know if there is any aspect of the work you would like us to elaborate on.
>
> We look forward to your reply and greatly appreciate your feedback.

---

> ### Author Response · Authors · 2025-08-08
>
> Dear Reviewer PFAD,
>
> There are only 24 hours left in the discussion period. We have done our best to address your questions in the rebuttal, but have not yet received your response. If you have any further questions or concerns, we would be happy to provide clarification.
>
> We look forward to your reply and greatly appreciate your feedback.

---

### Official Review · Reviewer_okF2 · 2025-07-03

**Clarity:** 4
**Significance:** 4
**Originality:** 3
**Rating:** 4
**Confidence:** 4

**Summary:**

This paper presents a novel method, Symmetric Multimodal Preference Optimization (SymMPO), to mitigate hallucinations in Multimodal Large Language Models (MLLMs). It is a symmetric preference learning formulation. This method leverages contrastive image pairs (m, m') along with their respective preferred responses (yw, y'w) to provide direct preference supervision. This symmetric design not only enhances the model's ability to make fine-grained visual discriminations but also ensures the theoretical integrity of the DPO objective by aligning with its standard derivation. Furthermore, the method is augmented with a preference margin consistency regularizer to quantitatively constrain the preference gap between symmetric pairs, enforcing a more precise alignment.

**Questions:**

1. Whether other models can be used to enhance pair screening and acquisition (Besides CLIP).
2. Which specific stage or component of the "Caption-Anchored Claim Extraction-and-Rewriting" pipeline (Section 4.1) is primarily responsible for Fine-Grained task performance deficit? Is the limitation rooted in the initial caption generation, which may lack fine-grained detail, or does it stem from the subsequent claim extraction and rewriting process?

**Ethical Concerns:**

["NO or VERY MINOR ethics concerns only"]

**Final Justification:**

The manuscript and the subsequent rebuttal provided detailed experiments and results, and the authors addressed my concerns. I look forward to further work in this direction.

**Limitations:**

Yes

**Quality:**

3

**Strengths And Weaknesses:**

The paper proposed SymMPO framework is not merely an incremental improvement but an original and elegant solution that restores theoretical consistency with the foundational DPO loss. This demonstrates a high degree of methodological Quality and Originality.
The paper introduces a novel symmetric optimization strategy and a preference margin consistency regularizer (LMargin), which together provide a more direct and quantitative supervisory signal for visual understanding.

This work framework necessitates generating a unique preferred response for every contrastive image, which introduces additional computational complexity and cost compared to methods that reuse the original response. And it relies on CLIP similarity, but this heuristic may not always produce optimally challenging pairs. But it difficult to isolate the true contribution of the SymMPO algorithm itself versus the quality of the input data.

---

> ### Author Rebuttal · Authors · 2025-07-30
>
> Dear Reviewer okF2,
>
> Thank you for recognizing our work. We greatly appreciate your insightful comments and suggestions. Below, we provide detailed answers to your questions:
>
> > **Question 1:** Whether other models can be used to enhance pair screening and acquisition (Besides CLIP).
>
> **Response:**
>
> Thank you for this insightful question. Yes, since the primary objective of our contrastive image construction is to pair images that are visually similar but have subtle differences, any model capable of capturing fine-grained visual features and discriminating between nuanced differences is a potential candidate for this task.
>
> For example, self-supervised vision models such as **DINO** [1] and **DINOv2** [2] could serve as highly effective alternatives to CLIP. While CLIP is designed to align visual and textual representations, models like DINOv2 are optimized particularly for extracting high-quality, fine-grained visual features through self-supervised learning. This specialization makes DINOv2 especially well-suited for identifying images with minor visual differences.
>
> Additionally, advanced image segmentation models, such as **SAM (Segment Anything Model)** [3] can segment objects or regions within an image, enabling region-level feature extraction. These features could then be used to pair images with specific region-level similarities.
>
> We acknowledge that while CLIP served as a robust and practical choice for our current implementation, investigating alternative feature extractors is an excellent suggestion for enhancing SymMPO. We will consider this direction in our future work.
>
>
>
> > **Question 2**: Which specific stage or component of the "Caption-Anchored Claim Extraction-and-Rewriting" pipeline (Section 4.1) is primarily responsible for Fine-Grained task performance deficit? Is the limitation rooted in the initial caption generation, which may lack fine-grained detail, or does it stem from the subsequent claim extraction and rewriting process?
>
> **Response:**
>
> Thank you for your thoughtful question. You are correct that the performance deficit on fine-grained tasks **primarily stems from the initial caption generation stage**, which serves as the foundational step in our "Caption-Anchored Claim Extraction-and-Rewriting" pipeline.
>
> Based on our extensive experiments, we found the caption generation process, powered by the Qwen2.5-VL-32B model, exhibits two key limitations that hinder fine-grained task performance:
>
> 1. Lack of Fine-Grained Detail: The prompt used in current pipeline is primarily designed to generate global scene descriptions, rather focusing on producing fine-grained details. As a result, the captions often miss critical nuances or specific information. This lack of detailed captions impacts the subsequent claim extraction process. As a result, the generated claims tend to be overly generic, lacking the specificity required for fine-grained tasks [4]. For example, if the model only describes the overall scene without recognizing detailed features such as the shape of a bird’s feet, wings or beak, it would struggle to accurately classify its species.
> 2. Propagation of Hallucinations: Hallucination remains a significant challenge in current MLLMs. The Qwen2.5-VL-32B model sometimes generates captions containing **hallucinations**. In these instances, the subsequent claim extraction process may inadvertently incorporate these inaccuracies into claims, leading to distorted or erroneous preference data.
>
> In summary, these limitations in the initial caption generation stage can trigger a **cascading effect** throughout the preference data construction pipeline, ultimately compromising the quality of the generated preference data, particularly for tasks requiring fine-grained recognition. In the future, we plan to explore solutions to address this challenge.
>
>
>
> **References:**
>
> [1] Caron, Mathilde, et al. "Emerging properties in self-supervised vision transformers." ICCV 2021.
>
> [2] Oquab, Maxime, et al. "Dinov2: Learning robust visual features without supervision." TMLR 2024.
>
> [3] Kirillov, Alexander, et al. "Segment anything." ICCV 2023.
>
> [4] Yucheng Shi, et, al.; “Enhancing Cognition and Explainability of Multimodal Foundation Models with Self-Synthesized Data”. ICLR 2025.

---

> > ### Comment · Reviewer_okF2 · 2025-08-05
> >
> > Thanks for the author's detailed explanation, I'm still concerned about whether the Caption-Anchored Claim Extraction-and-Rewriting process can completely remove the illusion or provide enough detailed information.
> >
> >  I would keep score unchanged.

---

> > > ### Author Response · Authors · 2025-08-05
> > >
> > > Dear Reviewer okF2,
> > >
> > > Thank you for your response and for taking the time to provide further feedback. We acknowledge your concern regarding the ability of the Caption-Anchored Claim Extraction-and-Rewriting process to completely eliminate hallucinations and ensure sufficient detail in the data.
> > >
> > > It is important to clarify that the **primary focus and contribution** of our work is on addressing the theoretical limitations of existing multimodal DPO methods. As such, we deliberately positioned the data construction pipeline within the **experiment section** rather than the **method section** to clarify that it is not intended as a core contribution of our work. Additionally, we explicitly acknowledge the limitations of the data pipeline in the **Section 4.3** to maintain transparency about its current shortcomings. The intent was to highlight that while the dataset quality is relevant, the novelty and contributions of our work lie in solving the theoretical challenges in multimodal preference learning.
> > >
> > > Furthermore, even within the existing data construction framework, our proposed **SymMPO** method consistently outperformed mDPO under the same experimental conditions. This demonstrates the effectiveness and robustness of our approach, independent of the limitations in the data construction pipeline.

---

> > > ### Author Response · Authors · 2025-08-07
> > >
> > > Dear Reviewer okF2,
> > >
> > > Thank you for the time and effort you have devoted to reviewing our paper!
> > >
> > > With the discussion period now extended, we would welcome the opportunity to clarify any remaining questions or concerns you might have. Please let us know if there is any aspect of the work you would like us to elaborate on.
> > >
> > > We look forward to your reply and greatly appreciate your feedback.

---

### Decision · Program_Chairs · 2025-09-17

**Decision:**

Accept (poster)

**Comment:**

This paper presents Symmetric Multimodal Preference Optimization (SymMPO), a new method to mitigate hallucinations in Multimodal Large Language Models (MLLMs). After rebuttal, it received scores of 4445. Generally, all the reviewers are positive about the paper, commenting that (1) the proposed SymMPO framework is novel and somewhat elegant, especially the design of the symmetric pairwise reward; (2) the motivation is reasonable and straightforward, the paper writing is good and easy to follow; and (3) experiments are comprehensive. Therefore, the AC would like to recommend acceptance.